# Diurnal Variation Characteristics of Summer Precipitation and Related Statistical Analysis in the Ili Region, Xinjiang, Northwest China

Zhiyi Li [1,†], Abuduwaili Abulikemu [1,*,†], Kefeng Zhu [2], Ali Mamtimin [3,4,5,6], Yong Zeng [3,7,8], Jiangang Li [3,7,8], Aerzuna Abulimiti [1], Zulipina Kadier [1], Abidan Abuduaini [1], Chunyang Li [1] and Qi Sun [1]

1   Xinjiang Key Laboratory of Oasis Ecology, College of Geography and Remote Sensing Sciences, Xinjiang University, Urumqi 830017, China; lizhiyi@stu.xju.edu.cn (Z.L.); aerzuna@stu.xju.edu.cn (A.A.); zulipinakadier@stu.xju.edu.cn (Z.K.); abidaabdugini@stu.xju.edu.cn (A.A.); chunyangli0326@stu.xju.edu.cn (C.L.); sunqi990129@stu.xju.edu.cn (Q.S.)
2   Key Laboratory of Transportation Meteorology of CMA, Nanjing Joint Institute for Atmospheric Sciences, Nanjing 210041, China; zhukf@cma.gov.cn
3   Institute of Desert Meteorology, CMA, Urumqi 830002, China; ali@idm.cn (A.M.); zengyong@idm.cn (Y.Z.); lijg@idm.cn (J.L.)
4   National Observation and Research Station of Desert Meteorology, Taklimakan Desert of Xinjiang, Urumqi 830002, China
5   Taklimakan Desert Meteorology Field Experiment Station of CMA, Urumqi 830002, China
6   Xinjiang Key Laboratory of Desert Meteorology and Sandstorm, Urumqi 830002, China
7   Field Scientific Observation Base of Cloud Precipitation Physics in West Tianshan Mountains, Urumqi 830002, China
8   Xinjiang Cloud Precipitation Physics and Cloud Water Resources Development Laboratory, Urumqi 830002, China
*   Correspondence: abduwaly@xju.edu.cn
†   These authors contributed equally to this work.

**Abstract:** The diurnal variation characteristics and basic statistical features of summer precipitation (from June to August) in the Ili region from 2015 to 2019 were investigated based on 4 km resolution Weather Research and Forecasting model simulation data from Nanjing University (WRF_NJU). The results show that the overall diurnal variation characteristics of precipitation (DVCP) reflected by the WRF_NJU data were consistent with respect to the observations and reanalysis data. The total precipitation pattern exhibited high (low) values on the east (west), with higher (lower) values over the mountainous (valley) area. Hourly precipitation amount (PA), precipitation frequency (PF), and precipitation intensity (PI) show similar diurnal variation characteristics, with peaks occurring at around 1700 LST in the mountainous area and around 2000 LST in valleys. Furthermore, moderate to intense precipitation contributes up to 87.88% of the total precipitation. The peaks in the mountainous area occur earlier than the valleys, while the peaks in western part of the valleys occur earlier than the eastern part. The PA peaks over the valleys and slopes occurred from the evening to early morning and from the afternoon to evening, respectively. In addition, the rotated empirical orthogonal function (REOF) analysis implied that the DVCP exhibits distinct differences between mountainous and valleys, and peak precipitation occurs during the evening in basin– and wedge–shaped areas, while the mountain peaks and foothill regions exhibit semi–diurnal variation characteristics. Among several basic meteorological factors, the vertical velocity (VV) and water vapor mixing ratio (WVMR) provided major contributions to the DVCP in both areas with high and low coefficients of variation, and the WVMR (VV) probably played a more significant role in mountainous (valleys) areas.

**Keywords:** diurnal variation; Ili region; precipitation evaluates; REOF; CV

## 1. Introduction

Precipitation is the result of complex processes in the atmosphere and is an important component of the global water cycle [1,2]. With global climate warming, there is an increasing trend in both the intensity and frequency of extreme precipitation events in many parts of the world [3,4]. The Fifth Assessment Report of the Intergovernmental Panel on Climate Change (IPCC) has repeatedly noted that knowledge of future climate change continues to be heavily influenced by uncertainties in precipitation estimates [5,6]. Intense precipitation often leads to disastrous events such as flash floods, landslides, mudflows, and dam failures, which have tremendous impacts on agriculture, transportation, the ecological environment, daily life, and the economy [7–9]. Thus, enhancing the understanding of the characteristics of occurrence, development, causal factors, and other tempo–spatial features of precipitation has become one of the hotspots and focal points in the field of meteorology and climatology, as well as in some other broader fields of geography [10–12].

The diurnal variation of precipitation (DVP), as an important regional weather and climate characteristic [13–16], is a periodic variation that is mainly induced by solar radiation directed toward the Earth's atmospheric system [11,17]. The diurnal variations of tempo–spatial distribution, occurrence and development characteristics, intensity, frequency, and duration of precipitation have significant impacts on ecological balance, water cycle, and human production and life [11,18]. Enhancing in–depth research on DVP has multiple benefits. On the one hand, it helps to accurately understand the tempo–spatial distribution characteristics and variation patterns of regional precipitation, and further comprehend the formation mechanisms of precipitation. On the other hand, it provides a scientific basis for evaluating numerical models and serves as a reference for improving the modeling and forecasting of regional precipitation [19,20].

Significant research achievements have been attained in the field of DVP. The studies generally employ natural units or administrative divisions to investigate diurnal variations from top to bottom [17,21]. Previous research studies were primarily conducted using ground–based observational data. Sen Roy and Balling [22] studied the diurnal variation characteristics of precipitation (DVCP) in India using data from 2000 stations. They found that different regions in India exhibited distinct characteristics in terms of precipitation frequency, peak timing, and the influence of coastal and inland locations as well as topography. In Europe, Twardosz et al. [1] studied the peak values and phases of precipitation frequency and total amount, revealing pronounced seasonal differences in the DVCP in southern Poland. In the United States, Wootten et al. [23] conducted statistical analyses using summer precipitation data in North Carolina, and they identified significant statistical differences between the daytime and evening precipitation amount (PA).

In a research study on DVP in Taiwan, it was found that precipitation amounts and peak timing exhibit significant east–west differences due to the influence of the Central Mountain Range [24]. Furthermore, studies on the diurnal variation and phase of daily precipitation peak timing in different regions of mainland China have revealed distinct local characteristics in terms of seasonality and regional variations [11]. In the central and eastern parts of China, which are primarily composed of plains and hills with relatively gentle terrain and automatic weather stations (AWSs), it was shown that summer precipitation in South China and Northeast China reaches its peak in the late afternoon [25–28]. In contrast, precipitation in Southwest China and North China reaches its peak around midnight [18,29,30], while in the middle and lower reaches of the Yangtze River, precipitation is mainly concentrated in the late afternoon [31,32]. In most areas of the Qinghai–Tibet Plateau, precipitation exhibits two diurnal peaks: one in the early morning (midnight) and the other in the late afternoon [33]. With the rapid development of high–tempo–spatial–resolution meteorological data, many scholars have conducted in–depth studies on the tempo–spatial distribution of DVCP in complex terrain areas [34]. Chen et al. [35] found that convective features appear most frequently in the southern coastal and eastern mountainous areas of the Pearl River Delta (PRD) region based on analyses of Doppler radar data for a duration of 3 years over the

PRD region, with the highest frequency in June and the lowest frequency in September. Liu et al. [36] used TRMM to analyze the hourly precipitation in the summer of 2008–2014 in this region, and found that the hourly precipitation and precipitation frequencies in the eastern and central regions were greater than those in the western regions. There are obvious differences between high precipitation intensity (PI) and high precipitation frequency (PF).

Due to the complex terrain and relatively sparsely distributed observation stations, previous studies on precipitation in Xinjiang have mainly focused on individual case studies, and some other long–term tempo–spatial distribution characteristics of precipitation [37–39]. Previous research on DVCP in Xinjiang mainly relied on station data, reanalysis data and satellite precipitation data with coarse spatial resolution (e.g., Cao et al. [40]). Li et al. [41] analyzed the DVCP and its influencing factors in the central Tianshan Mountains using satellite and AWS data. Jing et al. [42] conducted a statistical analysis of the main meteorological factors affecting winter precipitation in the Ili River Valley (IRV) using reanalysis data and AWS data. However, previous studies on DVCP in Xinjiang lacked high–tempo–spatial–resolution data to explore the DVCP in complex terrain areas during the summer, and the quantitative statistical relationship between small–scale meteorological factors, local terrain, and precipitation in Xinjiang have also not been explored to date.

The Ili region is located in the center of Eurasia continent, and surrounded by mountains (Tianshan Mountain Range) on three sides, with an open flat terrain on its western boundary, forming a trumpet–shaped terrain. The convergence and updraft of water vapor carried by the westerly flow make it the wettest region in Xinjiang, known as the "Central Asian Wet Island" [43]. With global climate warming, the tempo–spatial distribution characteristics of precipitation over the IRV exhibit significant inhomogeneity. Moreover, the region is considered to be one of the most frequent heavy rain areas in Xinjiang in summer [39], and exhibits significant diurnal variation characteristics influenced by the mountain–valley terrain, underlying surface, atmospheric circulation patterns, and thermodynamic mechanisms [12,44]. In addition, the extreme precipitation events in the Ili region also exhibited an increasing trend, for instance, a single rainstorm process on 16 June 2016, caused severe economic losses of nearly 600 million CNY, with the disaster–affected population exceeding 70,000 [37,45]. Previous studies on the DVCP in the Ili region relied mainly on observation data from a few scattered and sparsely distributed stations, which cannot accurately distinguish the surrounding areas with mountainous or other steep terrains. Therefore, previous findings can not be considered as representative of the surrounding areas of the stations, and they cannot provide a clear understanding of the distribution of precipitation and DVCP in the Ili region and surrounding mountainous regions.

However, high–tempo–spatial–resolution numerical model data are considered an important tool for in–depth research on DVCP characteristics, precipitation forecasting, and simulation [16,20,46,47]. The real–time forecasting model Advanced Research Weather Research and Forecast (WRF_ARW), improved by Nanjing University, has frequently been used by scholars in recent years in the study of summer precipitation in the central and eastern parts of China. Zhu et al. [48] compared the simulation performance of WRF_NJU with other global models for summer precipitation in China in 2013 and found that the WRF_NJU's performance was even better. Cai et al. [49] found that the WRF_NJU model could reasonably reproduce the diurnal cycle of summer precipitation, including peak timing, duration, and amplitude, in the Qinghai–Tibet Plateau and Sichuan Basin. In an evaluation of the summer precipitation forecast from WRF_NJU in Xinjiang, Xu and Ming [50] stated that the diurnal variation in the precipitation forecast from WRF_NJU exhibited a consistent trend with observations, with the peak values occurring at a similar time, and the WRF_NJU simulation performed well for the majority of precipitation events. Consequently, WRF_NJU simulation data can be considered reliable for further studies on summertime precipitation, with a high tempo–spatial resolution.

In this study, ground–based AWS data, Multi–Source Weighted–Ensemble Precipitation (MSWEP), and ERA5–Land were used to evaluate the ability of the hourly 4 km resolution WRF_NJU model to simulate precipitation in the Ili region (i.e., IRV and its nearby mountainous area). Then, the DVCP in the Ili regions was studied, and a preliminary investigation was conducted on the quantitative relationship between main meteorological factors influencing the occurrence and development of precipitation and DVCP as the first investigation of DVCP, based on hourly 4 km resolution data in Xinjiang. The aim of this paper is to fill the gap in the in–depth exploration of the DVCP in the IRV and nearby mountainous areas due to the lack of high–tempo–spatial–resolution data. It contributes to an improved understanding of the DVCP in the Ili region and provides important insights that can be used to further explore the diurnal variation mechanisms of precipitation over complex terrain conditions [2,51].

This paper is organized as follows. Section 2 provides an overview of the study area and describes the information regarding the data and methods used in this research. Section 3 evaluates the capability of the WRF_NJU data in the region and presents the results of the analysis of DVCP and explores the correlation between the elevation, major meteorological factors, and DVCP in the Ili region. Finally, Sections 4 and 5 present the discussions and conclusions of the study.

## 2. Data and Methodology

### 2.1. Dataset

This paper utilizes automatic weather station (AWS) data provided by the China Meteorological Administration for the period from 1 June to 31 August 2015–2019, totaling 460 days. Data–quality control work was strictly conducted, taking into account data continuity, consistency, repetition rate, and missing rate [52]. The distribution of the stations is shown in Figure 1a. Besides, the Multi–Source Weighted–Ensemble Precipitation (MSWEP) V2 3–hourly 0.1° resolution dataset was used, which combines various remote sensing precipitation data, station data, and reanalysis data to obtain the highest quality precipitation estimates for each location [53,54]. This dataset has been widely used in studies on regional precipitation characteristics [55]. The ERA5–Land hourly 0.1° resolution precipitation data from the European Centre for Medium–Range Weather Forecasts (ECMWF) were also used [56]. They performed well in the DVCP studies over mountainous regions with complex topography [57].

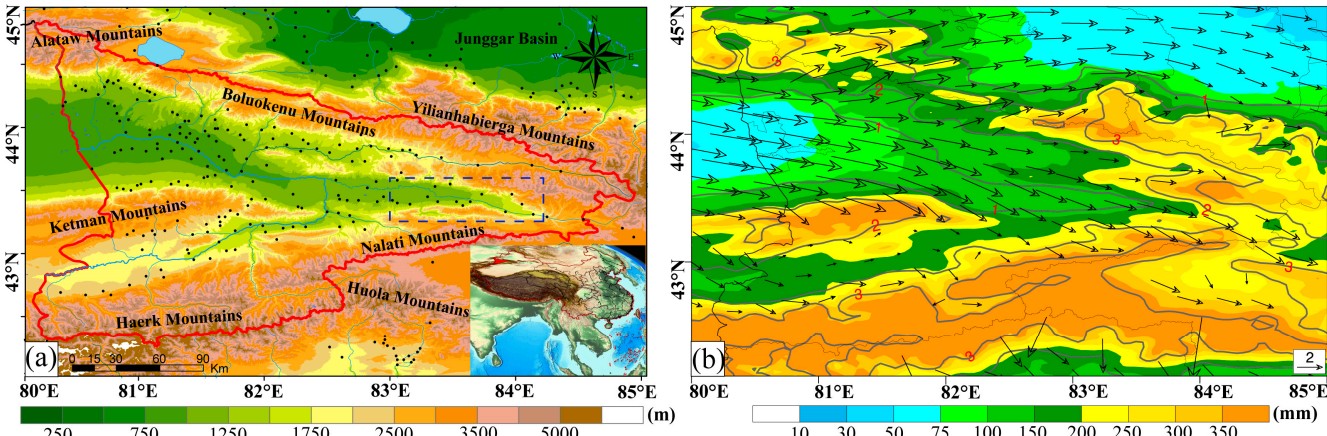

**Figure 1.** (**a**) Overview of the study area: the red line represents the administrative boundary of Ili region, the shaded area represents the topography, black dots represent automatic weather stations, and the blue lines represent rivers in the Ili region; the blue dashed box is an edge–shaped area. The inset map in the bottom right corner shows the location of the study area in China. (**b**) Summer average accumulated precipitation from June to August during the period of 2015–2019: the shaded colors represent precipitation, overlaid with the same averaged horizontal wind fields (arrows, units: m s$^{-1}$) at 750 hPa, and the topography is depicted by gray contour lines.

The 4 km resolution WRF_NJU forecast system has been providing real–time forecasts for China twice daily during the summer since 2013 [48]. The model domain has $1409 \times 1081$ horizontal grid points at a 4 km grid spacing with 51 vertical levels. Pivotal physics schemes used include the Morrison two–moment microphysics [58], the CAM short– and long–wave radiation schemes [59], the Pleim–Xiu land surface and surface layer schemes [60], and the Asymmetrical Convective Model Version Two Planetary Boundary Layer Scheme [61]. Considering the low quality of the simulation results during the model's spin–up period, the results of the 12–36 h forecast period were selected for the next day in each day's simulation.

The AWS data were used to evaluate the performance of the WRF_NJU data in flat areas below 1.5 km altitude (above sea level), while the MSWEP data were used to validate the performance of the WRF_NJU data in the entire study area, especially in regions lacking AWSs. The ERA5–Land were used to evaluate the precipitation simulation performance of the WRF_NJU under different terrain conditions.

### 2.2. Methods

The Coefficient of Variation (*CV*), represents the degree of variability of precipitation and is used to analyze the stability of precipitation over different time periods. A larger *CV* indicates a greater variation in precipitation over time, which increases the chances of droughts and heavy rainfall. Conversely, a smaller *CV* indicates better stability in precipitation. The formula used to calculate the *CV* is as follows:

$$CV = \frac{\sigma}{\overline{x}} \times 100\%, \tag{1}$$

$\sigma$ represents the standard deviation of hourly precipitation, and $\overline{x}$ represents the mean of hourly precipitation.

Rotated Empirical Orthogonal Function (REOF) is often used to study possible spatial modes (ie, patterns) of variability and how they change over time in climatology. REOF can effectively capture local features within a region and reveal the interrelationships among different areas. Additionally, REOF tends to have reduced sampling errors. Its significance testing is often performed using the North test [62]; for the derivation of the formulas, please refer to the referenced literature [63,64].

The Correlation Coefficient (*CC*) is a statistical measure used in meteorology and climatology to assess the degree of correlation between station–based observed precipitation, blended precipitation data from multiple sources, and reanalysis or forecast data. A higher absolute value of the *CC* indicates a stronger correlation between the datasets. The results are often subjected to an R correlation test; the formula for calculating the *CC* is as follows:

$$CC = \frac{\sum (x_i - \overline{x})(y_i - \overline{y})}{\sqrt{\sum (x_i - \overline{x})^2 (y_i - \overline{y})^2}}, \tag{2}$$

Root Mean Square Error (*RMSE*) is a meteorological statistical measure used to quantify the deviation between reanalysis data, forecast data, and observed precipitation. It is sensitive to outliers or extreme values in the data. A smaller *RMSE* value indicates a smaller deviation or error between the analyzed or forecasted precipitation and the observed precipitation [65]. The formula for *RMSE* is as follows:

$$RMSE = \sqrt{\frac{1}{n} \sum_{i=1}^{n} (x_i - y_i)^2}, \tag{3}$$

The relative bias (*BIAS*) is used to reflect the relative deviation between reanalysis and forecast data compared to observed data. It quantifies the degree of relative deviation between these datasets. The equation for calculating *BIAS* is as follows:

$$BIAS = \frac{\sum\limits_{i=1}^{n}(x_i - y_i)}{\sum\limits_{i=1}^{n} y_i} \times 100\%, \tag{4}$$

In the equation, $x_i$ represents the reanalysis data or forecast data, and $y_i$ represents the observed data or merged precipitation data from multiple sources.

Hourly average precipitation amount (PA):

$$PA = \sum\limits_{day=1}^{n} prcp(hour, day)/n, \tag{5}$$

Hourly average precipitation frequency (PF):

$$PF = \sum\limits_{day=1}^{n} pf(hour, day)/n \times 100\%, \tag{6}$$

Hourly average precipitation intensity (PI):

$$PI = \frac{PA}{PF}, \tag{7}$$

In the equation, $prcp(hour, day)$ represents the PA at every hour of each day. $pf(hour, day)$ represents the number of occurrences for the precipitation at each hour is greater than or equal to 0.01 mm. PI represents the total PA at a specific moment divided by the total number of occurrences of precipitation at that moment.

Taking full account of the geographical location of Ili region, the time division is as follows (Table 1):

**Table 1.** Time slot names and their corresponding time ranges used for DVCP analysis in this paper.

| Time Slot Name | Time Range (LST = UTC + 6) |
|---|---|
| Midnight | 2300–0100 |
| Early morning | 0200–0400 |
| Dawn | 0500–0700 |
| Morning | 0800–1000 |
| Noon | 1100–1300 |
| Afternoon | 1400–1600 |
| Nightfall | 1700–1900 |
| Evening | 2000–2200 |

## 3. Results

### 3.1. Precipitation Data Evaluation

Observational and gridded data are often used to evaluate the weather and climate models. However, due to the high tempo–spatial variability in precipitation and measurement errors, precipitation data are prone to errors. Previous studies have found that, in mountainous areas, data are scarce, and this can cause errors of about 60% [66]. Therefore, multiple precipitation datasets from different sources (such as AWS, MSWEP, and ERA5–land) were considered to evaluate the WRF_NJU data in this study. At the same time, we also selected AWS data in the station's relatively densely distributed areas (valley plain area below 1.5 km ASL) to minimize the impact

of under–sampling precipitation. We used precipitation averages and larger scale statistics to increase confidence.

Figure 2a shows the temporal distribution of four sets of precipitation data. From the graph, it can be seen that the hourly precipitation in the Ili region exhibits a clear unimodal structure, which is consistent with previous studies on diurnal peaks in summer precipitation [67]. They utilized hourly precipitation data from three meteorological stations in western Ili and one station in eastern Ili to study the DVCP and found that the maximum precipitation occurred at 2000 LST and the minimum precipitation occurred at 1100 LST. By employing the Cressman interpolation method to interpolate the data from AWSs across the entire study area, it was discovered that the peak precipitation for the entire study area was 0.125 mm h$^{-1}$ at 2000 LST, and the minimum value was 0.058 mm h$^{-1}$ at 1200 LST. This finding is consistent with the results obtained by Yang et al. [68], who conducted a study using data from 10 meteorological stations in the Ili River Valley (IRV). They found that summer precipitation exhibited an unimodal pattern, with the peak occurring at 2000 LST. On the other hand, the peak precipitation values for MSWEP, WRF_NJU, and ERA5–Land were 0.132 mm h$^{-1}$, 0.164 mm h$^{-1}$, and 0.297 mm h$^{-1}$, respectively, appearing at 1600 LST, 1700 LST, and 1600 LST. The minimum precipitation values were 0.049 mm h$^{-1}$, 0.063 mm h$^{-1}$, and 0.086 mm h$^{-1}$, respectively, occurring between 0200 and 0600 LST. From Figure 1a, it can be seen that the AWSs are mainly located in flat IRV below an altitude of 1.5 km. Therefore, for the evaluation of precipitation data in the entire study area, MSWEP data were used as the reference for comparison. Figure 2b shows the standard deviation (SD), correlation coefficient (*CC*), root mean square error (*RMSE*), and relative bias (*BIAS*) between WRF_NJU, ERA5–Land, and MSWEP as 1.04, 2.36, 0.89, 0.99, 0.03, 0.08, 33.55%, and 93.08%, respectively. Taking full account of the geographical location of the Ili region, the time division is as follows. It can be seen that, for the temporal variation in precipitation in the entire study area, WRF_NJU is close in magnitude to the reference data MSWEP and exhibits a consistent temporal trend. This is similar to previous evaluation results where the diurnal variation and peak values of WRF_NJU precipitation data closely align with the observations [50]. Due to the limitations regarding the number and spatial distribution of AWSs, the data from the stations located below 1.5 km in altitude and on flat terrain were used as a reference. Similarly, the WRF_NJU precipitation values in Figure 2a are close to the observations, and the trends are relatively consistent. Figure 2b reveals that the SD, *CC*, *RMSE*, and *BIAS* are 1.17, 0.65, 0.01, and −0.6%, respectively, which are significantly better than the results from the other three datasets.

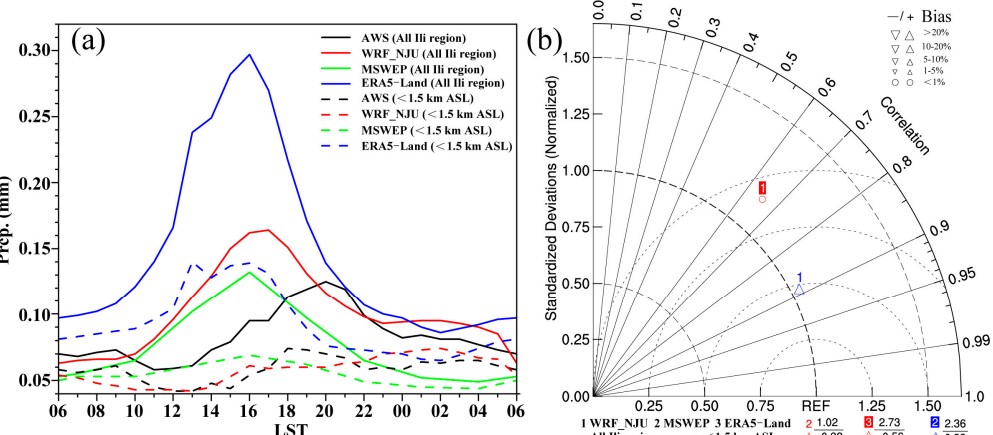

**Figure 2.** (**a**) The diurnal variation curve of average hourly characteristics of precipitation during summer. (**b**) The Taylor diagram illustrates the comparison between WRF_NJU data and ERA5–Land with AWS and MSWEP. The red color indicates the comparison between reanalysis data and forecast data below 1.5 km with the data from AWSs. The blue color represents the comparison between WRF_NJU data and ERA5–Land within the entire study area with MSWEP.

Figure 3 presents four sets of precipitation data (0200, 0800, 1400, 2000 LST) depicting the PA levels in the IRV and its surrounding mountainous areas. The areas above an elevation of 1.5 km use the MSWEP precipitation data as a reference, while the areas below 1.5 km use data from AWSs as a reference. A comparison is made between the WRF_NJU model and ERA5–Land data. Based on the four sets of precipitation data and time division method (Table 1), it can be found that, in the morning, at 0800 LST, there is a strong distribution of precipitation in the southern mountainous region of the IRV, with precipitation amounts ranging from 0.2 to 0.3 mm. The lowest precipitation values are found in the northwest part of the study area, with an average precipitation amount of less than 0.05 mm (Figure 3a–d). In the afternoon at 1400 LST, the precipitation range expands in the mountainous areas near the IRV, and the magnitude increases. The precipitation pattern is consistent with the orientation of the mountains, with a northwest–southeast trend in the northern part and a southwest–northeast trend in the central and southern parts. The highest precipitation occurs near the Ketman Mountain in the central region, while the lowest precipitation distribution range was found in the northwest valley, which expands towards the east (Figure 3e–h). In comparison to the earlier time period, the precipitation range and magnitude in the mountainous regions diminish in the evening at 2000 LST, especially in the northern mountainous region. The precipitation amount increases within the valley area (Figure 3i–l). In the early morning, at 0200 LST, the precipitation range and magnitude in the mountainous areas further diminish, while the lower–altitude valley area experienced an expansion in the precipitation range. Through a comparison of the observational data and the reanalysis data, it can be concluded that the WRF_NJU model accurately simulated the temporal distribution characteristics of precipitation in the IRV and its surrounding mountainous areas. Therefore, these data can be considered credible, and used for future studies on the DVP and related meteorological factors within the research area (Figure 3m–p).

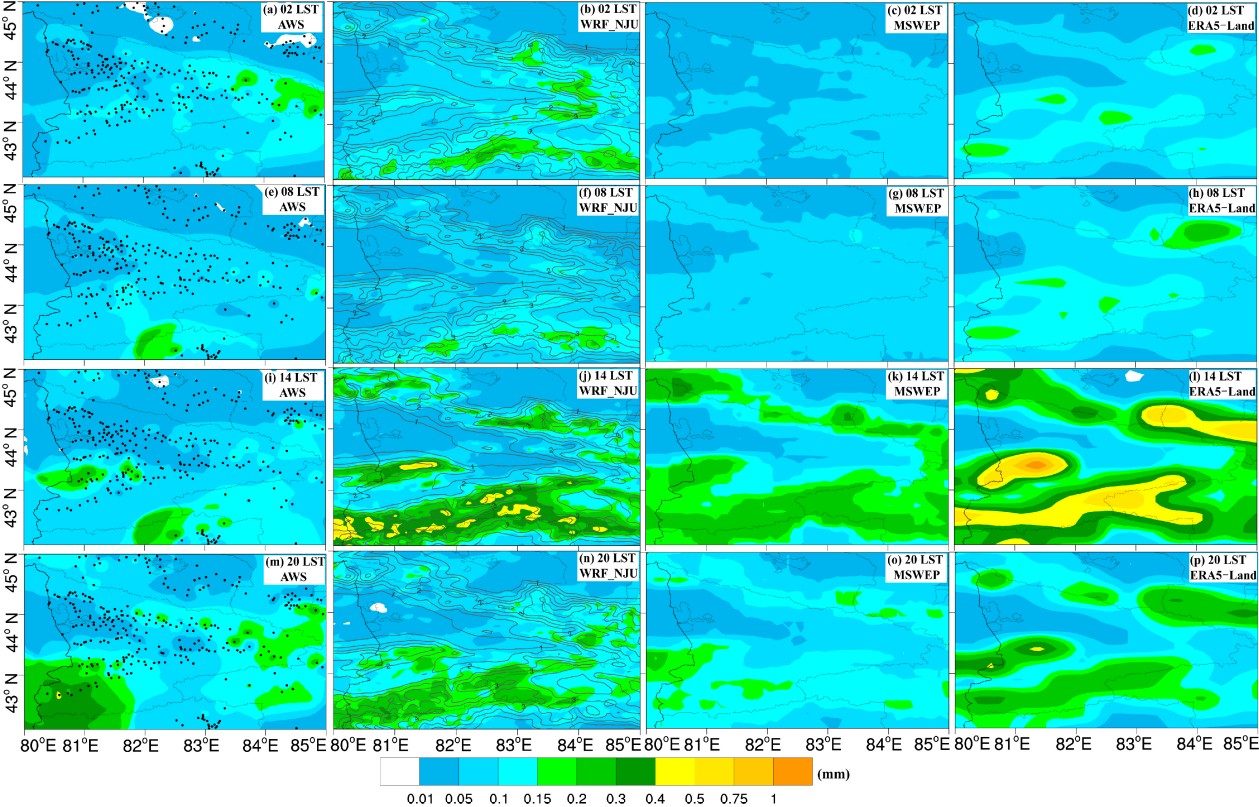

**Figure 3.** The spatial distribution of hourly average precipitation during summer, every 6 h from 02 to 20 LST. The black lines represent administrative boundaries, the gray lines represent topography, and the black dots represent AWSs. The shading represents the hourly average precipitation (mm).

*3.2. The Diurnal Variation Characteristics of Precipitation in the Ili River Valley*

3.2.1. The Diurnal Variation of Precipitation Frequency (PF) and Precipitation Intensity (PI)

Because Xinjiang has a much lower density of distribution of conventional and automatic weather stations than the mid–eastern part of China due to its sparse population, along with a slower economic development, complex topography, and harsh environment such as the desert and gobi, the WRF_NJU data with relatively high tempo–spatial resolution can offer a good opportunity to investigate much more detailed characteristics of DVCP in Xinjiang.

It can be seen from Figure 4 that the spatial distribution characteristics of PF at 0200 LST in the early morning are similar to those of PA (Figure 2a). The overall PF ranges from 10% to 20%, with southern regions having higher values than northern regions, and eastern regions having higher values than western regions. The maximum PF is located in the southern mountainous area, whereas the minimum PF is found in the northwest valley of the study area (Figure 4a). At 0500 LST in the dawn, little has changed in PF. The PF range slightly widens in the southern portion of the valley while it decreases in the northern part (Figure 4b). By 0800 LST in the morning, the PF significantly decreases. With the exception of the southern mountainous area, where the frequency typically falls below 10% at this time, there is a substantial decrease in the occurrences of precipitation across the study area. (Figure 4c). From 1100 LST at noon to 1700 LST at nightfall, the PF increases significantly, consistent with the trend shown in the precipitation distribution map. The southern mountainous area exhibits a clear band–shaped pattern, with PF increasing from 20% to over 60%. The eastern part has a higher frequency compared to the western part, while the magnitude and range of PF within the valley do not change significantly, remaining below 20%. Additionally, it is evident that the PF in the northern mountainous area is below 40%, lower than in the central and southern mountainous areas, with a small range of minimum PF occurring in the Boluokenu Mountains (Figure 4d–f). From 2000 LST in the evening to 2300 LST at midnight, the PF gradually decreases to below 20% in the northern mountainous area, and the central and southern mountainous areas also show a reduced frequency and range. Conversely, within the valley, there is a slight increase in the magnitude and range of PF. The PF in the IRV and its surrounding areas shows distinct diurnal variations. The highest PF, exceeding 60%, occurs around 1700 LST, while a small region of the western Ili River basin near (44°N, 80.5°E) contains areas with a PF below 5%. Overall, the lowest PF occurs at around 0800 LST.

Similar to PA and PF, the PI in the nearby mountainous areas is generally higher than 0.5 mm h$^{-1}$ compared to the valley areas. The PI gradually increases in the western, central and southern parts of the northern mountainous area from 0500 to 1400 LST (Figure 5b–e), reaches its maximum value around nightfall, near 1700 LST, and gradually decreases thereafter (Figure 5f). The maximum PI in the western mountainous area exceeds 2 mm h$^{-1}$ and occurs during the evening period, from 2300 to 0500 LST (Figure 5a,b,h), while it is relatively weaker during the daytime. In the valleys, the evening PI is greater than during the daytime. In the western part, the PI exhibits larger daily variations, ranging from 0.01 to 0.75 mm h$^{-1}$, compared to the eastern part, which ranges from 0.25 to 0.75 mm h$^{-1}$. In the northern mountainous area, the maximum PI in the eastern section occurs at around 1100 LST (Figure 5d). Yang et al. [68] investigated the hourly average PI and shows significant differences in the timing of peak PI based on observations from five meteorological stations in the western part of the IRV, which indicate that the peak in PI occurrs between 18:00 and 19:00 LST. However, observations from several stations cannot solely accurately reflect the tempo–spatial distribution characteristics of precipitation in the region, as the data are not representative of the entire or wider region.

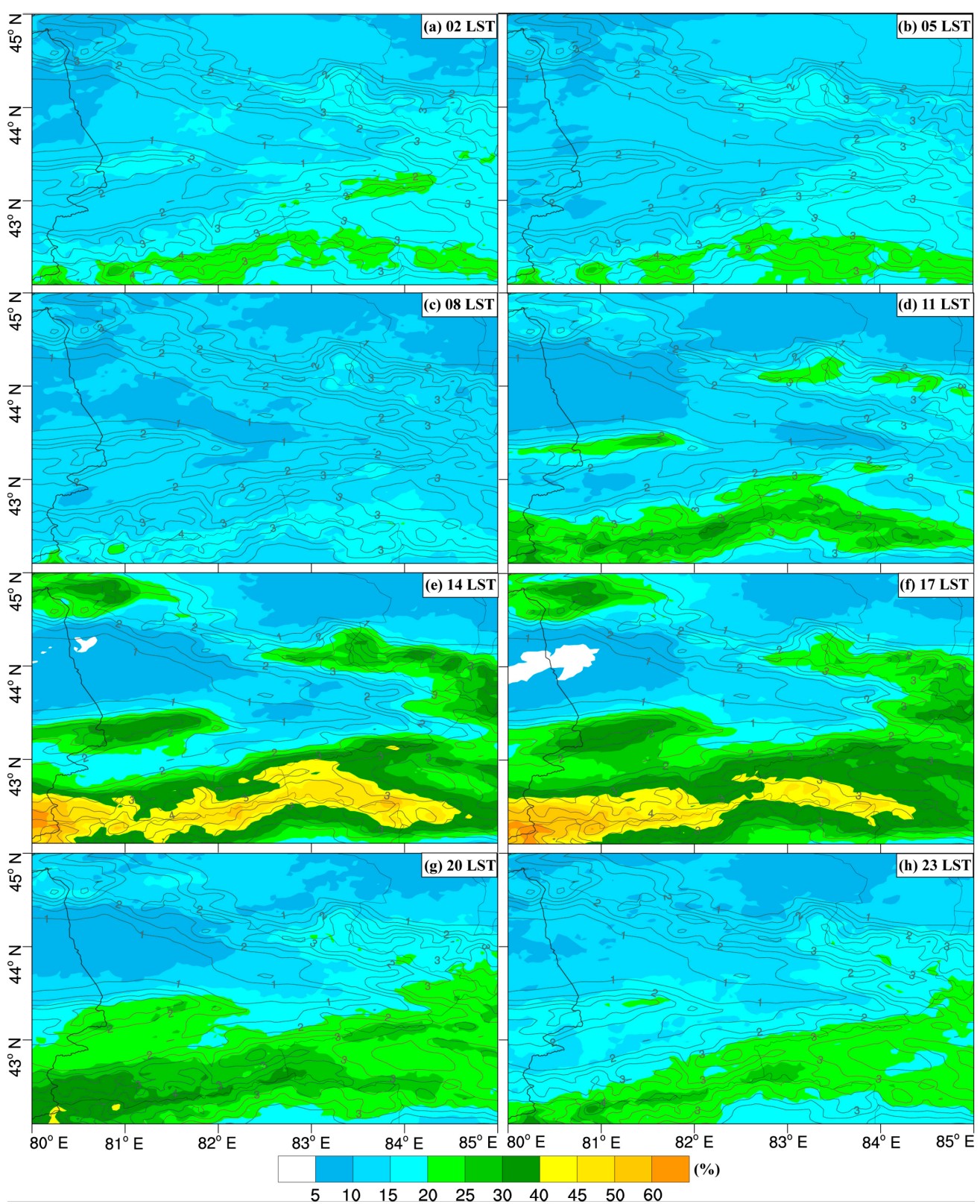

**Figure 4.** Spatial distribution of hourly averaged precipitation frequency (PF) every 3 h, from 02 to 23 LST, during the summer (from June to August) in the Ili region from 2015 to 2019. Gray lines represent topography, and color shading indicates the PF (unit: %).

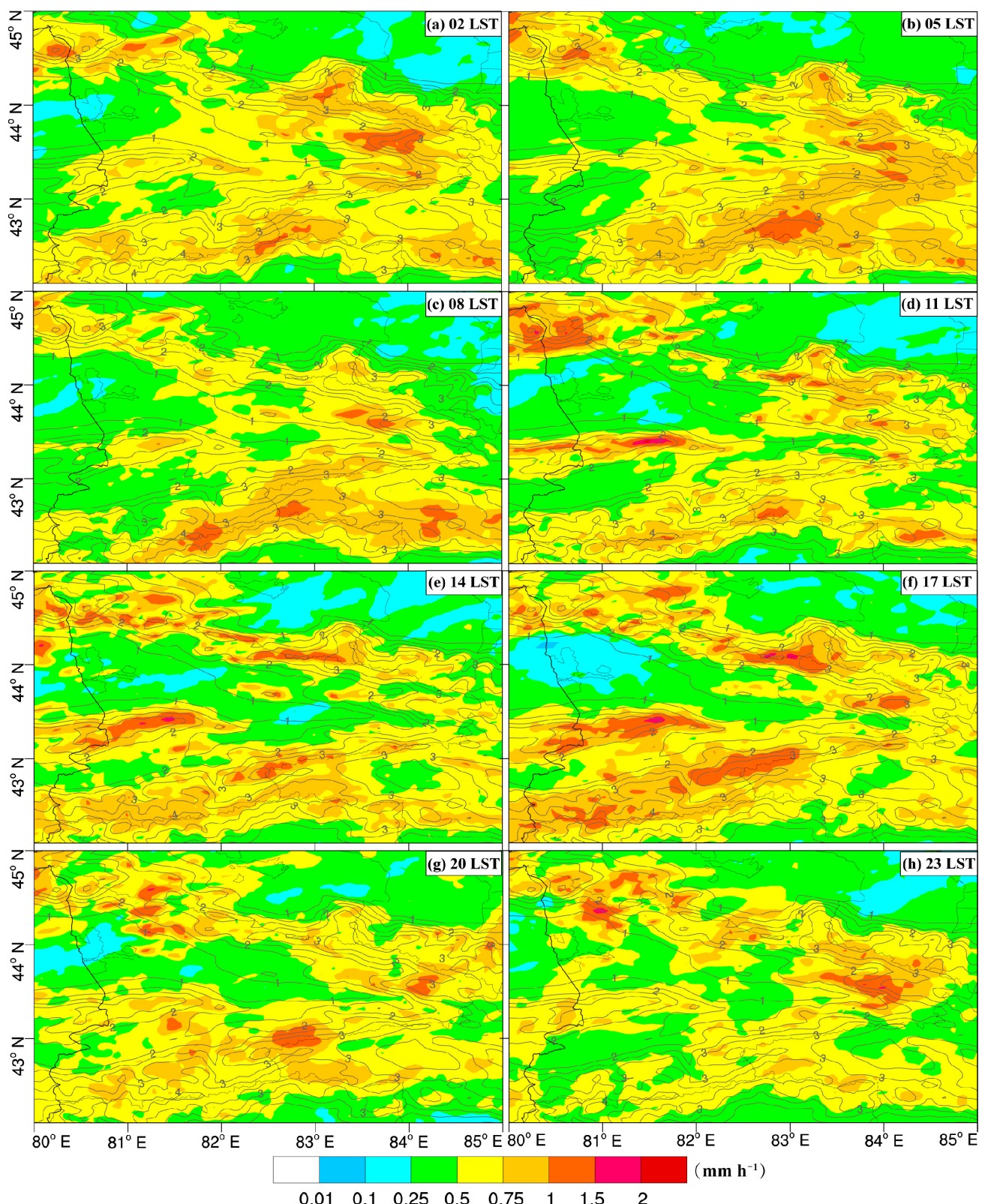

**Figure 5.** Spatial distribution of hourly averaged precipitation intensity (PI) every 3 h, from 02 to 23 LST, during the summer (from June to August) in the Ili region from 2015 to 2019. Gray lines represent topography, and color shading represents PI (unit: mm h$^{-1}$).

According to previous studies, DVP is usually determined by PF and PI [69,70]. In order to further investigate the diurnal variation characteristics of PI in the IRV and its adjacent mountainous areas, the hourly PI is divided into 10 categories, following the method of Karl and Knight [70]. It can be seen from Figure 6 that moderate and heavy precipitation contribute to over 80% of the total precipitation in each time period. Combining the characteristics of the PI in Figure 5, the frequency of occurrence and the contribution rate of moderate and heavy precipitation levels are higher during the midday–to–evening periods than during the evening–to–morning periods. PI exceeds 90% at 1500 LST, and contributes to approximately 50% of the total precipitation. This indicates that there are differences in the diurnal variation at different PI levels, with moderate and heavy precipitation being predominant. To further quantitatively analyze the contribution of the different PI levels to the total precipitation in the summer in the IRV, Table 2 shows that the accumulative hourly precipitation of PI levels 5–10 accounts for 87.88% of the total precipitation, with an average PI of 1.62 mm h$^{-1}$. The PI level 10 has a contribution rate of 27.25%, which is the highest among the 10 PI levels. In accordance with other research that discovered a high frequency of heavy precipitation in the mountainous regions of the IRV using case data (Figure 5), extreme heavy precipitation primarily occurs in steep mountainous areas and at the confluence of valleys [39,71].

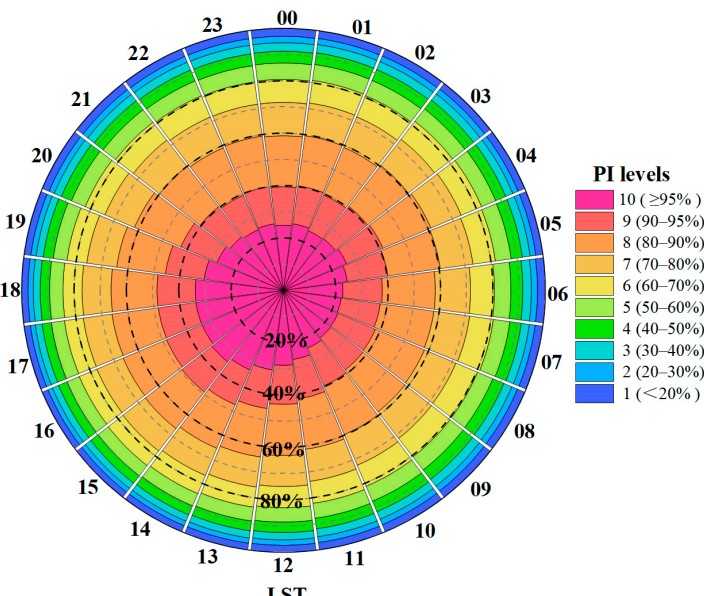

**Figure 6.** During summer (from June to August) in the Ili region from 2015 to 2019, obtained from WRF_NJU data. The percentage of total precipitation for each time period contributed by different PI levels of precipitation (unit: %).

**Table 2.** Contribution rate (unit: %) and average precipitation intensity (PI) (units: mm h$^{-1}$) of different intensity levels of summer rainfall in the Ili region.

| PI Levels | 5 | 6 | 7 | 8 | 9 | 10 | 5–10 |
|---|---|---|---|---|---|---|---|
| PI (mm h$^{-1}$) | 0.48 | 0.68 | 0.98 | 1.48 | 2.48 | 3.62 | 1.62 |
| Contribution rate (%) | 5.82 | 8.32 | 12.25 | 19.32 | 14.93 | 27.25 | 87.88 |

### 3.2.2. Diurnal Variation in Precipitation Based on REOF and *CV* Analysis

In order to clearly represent the DVCP under complex topographic conditions, a REOF analysis was employed to provide a tempo–spatial decomposition of the characteristics of PA. The North test was also employed for a significance analysis. Following the REOF decomposition, the cumulative contribution rate of the first three modes was achieved at 85.2% (Figure 7), and all three modes met the requirements of the significance test.

The first mode, which was substantially more dominant than the other modes and had a variance contribution rate of 61.2%, was able to explain the tempo–spatial distribution characteristics of precipitation in the Ili region, as well as a significant portion of the precipitation distribution. In the central and southern mountainous regions, the first mode's spatial distribution showed the presence of numerous high–value centers. Negative value areas were found along the valleys in a northwest–southeast direction, indicating variations in precipitation amounts with changes in terrain height. Negative value areas were found along the valleys in a northwest–southeast direction, indicating differences in precipitation amounts with changes in terrain height. The northern mountainous region, central and southern mountainous regions correspond to areas of high precipitation. Due to the presence of the Tianshan mountain range, the mountainous areas experience consistent precipitation characteristics. The areas with negative values correspond to the regions with low precipitation values in the IRV (Figure 7a). The second mode accounted for a variance contribution rate of 16.2%, with positive precipitation centers found in the Zhaosu Basin and near the vertex region of the wedge–shaped area, while a low–value center was found near the peak of Ketman Mountain (Figure 7b). The third mode accounted for a variance contribution rate of 7.8%, and the high–value center corresponded to the low–value area in the second mode (Figure 7c). According to a study on PA, PF, and PI, described above, the results showed that the highest values of precipitation in mountainous areas were primarily found between noon and 1200 LST and around nightfall, at 2000 LST, with the maximum value being attained at 1600 LST. According to Figure 7d, the peak precipitation in the valleys was between 2100 and 1100 LST in the evening, which is consistent with the characteristics of diurnal variations in precipitation noted in previous studies in the IRV [68]. The second mode captured the tempo–spatial pattern of a peak precipitation period from nightfall 1700 LST to early morning 0200 LST in the Zhaosu Basin and Labakou terrain area, showing differences from the mountainous and valley areas (Figure 7e). In contrast, the third mode exhibited the presence of two maximum peaks and two minimum peaks within the diurnal cycle (Figure 7f), which can be attributed to a mode related to the semidiurnal variation [24].

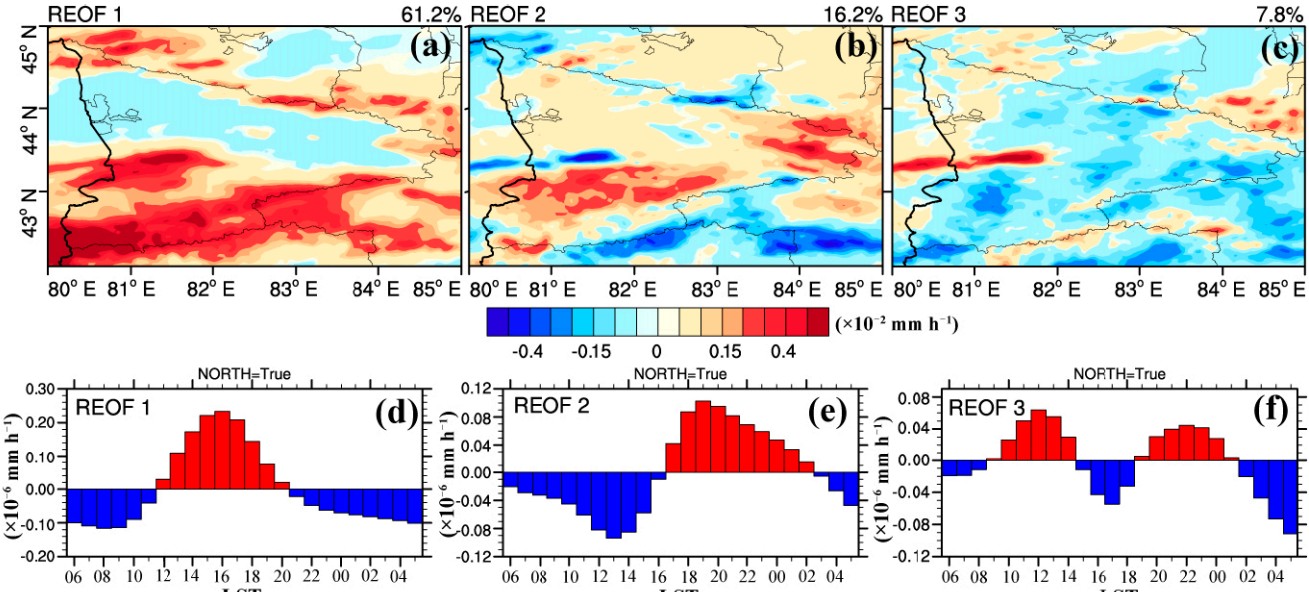

**Figure 7.** During summer (from June to August) in Ili region from 2015 to 2019 obtained from WRF_NJU data. REOF analysis of hourly precipitation in the Ili region during the summer season. Distribution of feature vectors for the first three dominant modes, (**a**) REOF1; (**b**) REOF2; (**c**) REOF3. (**d**–**f**) Amplitude of the weight time series is shown in the bottom panels.

To further elucidate the spatial distribution of abnormal DVCP, as shown in Figure 8, the *CV* was calculated for the PA in the IRV and adjacent mountainous areas, as shown in Figure 8. It can be seen that the *CV* of precipitation is higher in the western region compared to the eastern region. The western parts of the northern mountainous areas, the western parts of the IRV, the central mountainous areas, and southern mountainous areas all exhibit *CV* values exceeding 60%, with the maximum variation center located in the northwestern and southwestern parts of the study area exceeding 80%. In contrast, the *CV* in the eastern region, except for the southeastern mountainous area, is generally below 40%. Previous studies have found that high–value centers of *CV* reflect the degree of precipitation variability, corresponding to regions with significant differences in diurnal precipitation and frequent occurrences of heavy rainfall disasters [72]. According to the statistical research of Zhang et al. [73] on heavy rainfall events in Xinjiang from 1960 to 2018, the mountainous areas in the Ili region accounted for over 95% of the total strong precipitation events. Based on the previous analysis, we can infer that the IRV and nearby mountainous areas exhibit significant tempo–spatial differences in DVCP, which is consistent with the findings of Li et al. [74] using hourly precipitation data from automatic observation stations. The IRV shows significant spatial variability in summer PA and PF along the valley and towards the mountainous areas.

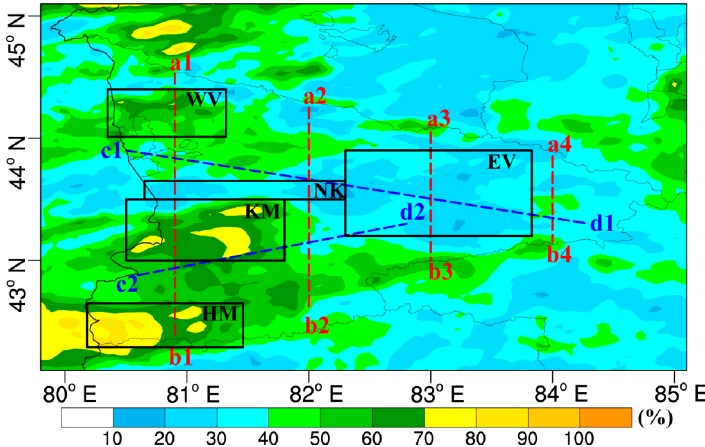

**Figure 8.** During summer (from June to August) in Ili region from 2015 to 2019, obtained from WRF_NJU data. Spatial distribution of the coefficient of variation for summer hourly average precipitation. The color shading represents the coefficient of variation (unit: %). The red (a1–b4) and blue (c1–d2) dashed lines indicate the locations of the vertical profiles in Figure 10. The black boxes represent the positions of the sub–regions in Figures 13 and 14.

Based on the results of the analysis of PA, PF, PI, REOF, and *CV*, five sub–regions were selected for further investigation (Figure 8): Western Valley (WV), Eastern Valley (EV), Ketman Mountains (KM), Northern Slope of Ketman (NK), and Haerk Mountains (HM). These sub–regions will be used for further regional analysis in the following sections.

### 3.2.3. Spatial Distribution Characteristics of Diurnal Variations in Precipitation

Figure 9a shows the spatial distribution of the diurnal variations in the peak time of PA obtained from the WRF_NJU data. To highlight the spatial distribution characteristics of the peak precipitation occurrence time, the precipitation data are masked by administrative boundaries. From Figure 9a, it can be seen that the peak precipitation shows clear north–south differences. The peak precipitation in the northern part occurs from evening to noon, while in the southern part, it occurs from noon to evening. Figure 9a shows that the peak values in the western part of the valley mostly occur from 2000 to 2200 LST, gradually shifting to the eastern part from 0800 to 1000 LST in the morning. It is evident that there is a gradual delay in the occurrence of peak precipitation. This result is similar to the findings of Li et al. [74] regarding the eastward movement of precipitation in summer regional

rainfall events in the IRV. Moving from the valley towards the mountains in the north and south, it can be shown that the peak precipitation in the mountainous areas generally occurs from 1400 LST to around 1900 LST, and there is also a time lag in the occurrence of peak precipitation from the mountains to the valley.

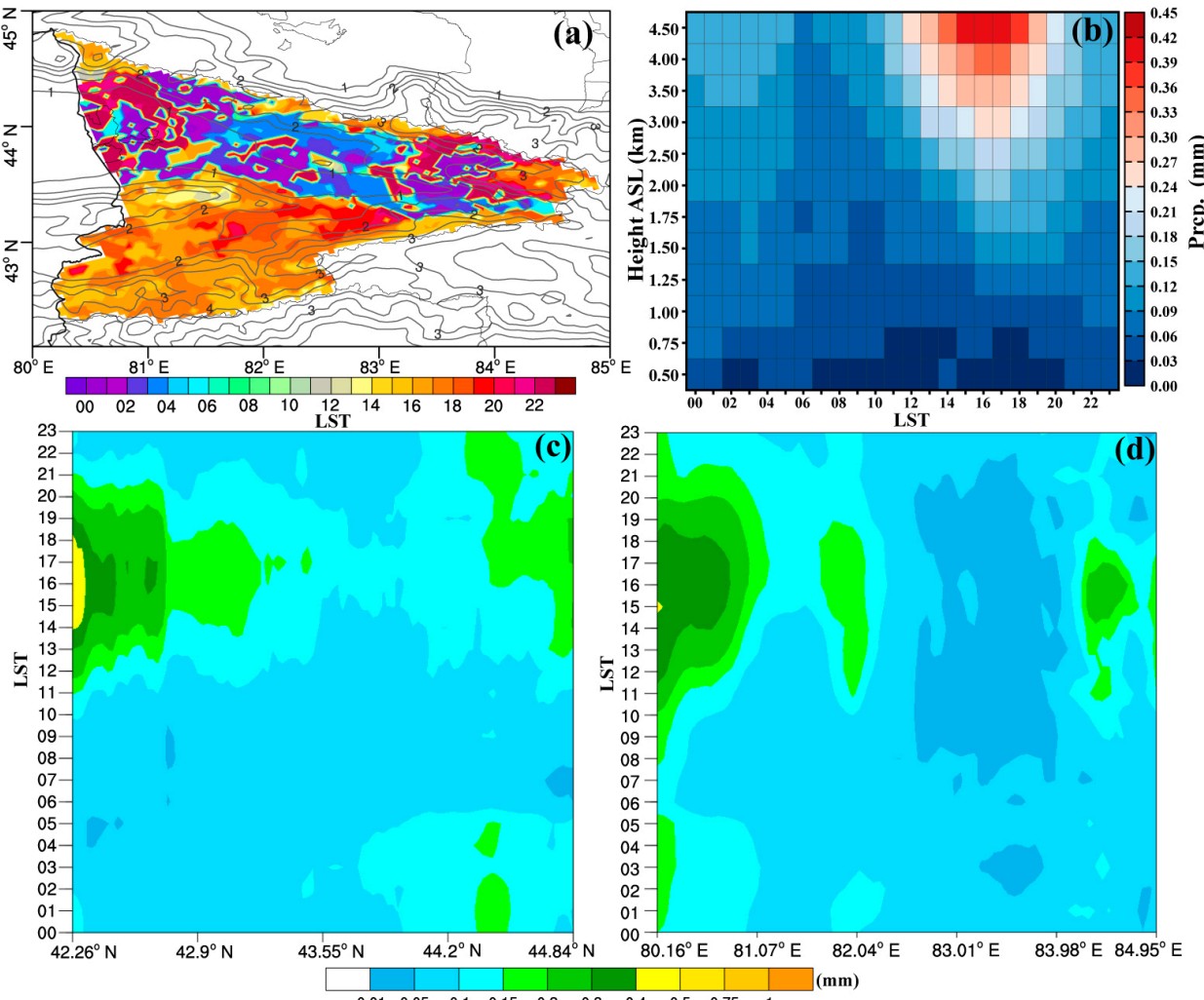

**Figure 9.** During summer (from June to August) in Ili region from 2015 to 2019, obtained from WRF_NJU data. (**a**) Spatial distribution of diurnal variations in hourly precipitation peak time (LST) in the Ili region during summer, with color representing time and gray contour lines representing topography. (**b**) Diurnal variations in hourly precipitation distribution at different elevations. (**c**) Time–longitude and (**d**) time–latitude hovmöller diagram of average hourly precipitation in summer.

Previous studies have found that, among the various topographical factors, elevation has a significant impact on the spatial distribution of climatological precipitation [11]. In order to investigate the characteristics of precipitation variations with altitude under complex terrain conditions in the Ili region, the study fully considers the topographic features of the research area in Figure 9b. The altitude is divided into intervals that are 0.5 km above and 0.25 km below. The corresponding PA is calculated for each height range. The results show that precipitation generally increases with increasing elevation, and there are significant differences in DVP at different elevations. The maximum precipitation occurs at around 1600 LST in the mountainous areas above 4 km, and the difference in DVP in the mountainous areas is significantly larger than that in the IRV at lower elevations. The peak precipitation in the IRV at lower elevations is concentrated from the evening to the early morning period.

The averaging is carried out along the meridional direction using the boundaries of the Ili region (Figure 9c), taking into account the topographic profile of the research area, which is wider in the west and a narrower in the east. It can be seen that there is a significant spatial variation in precipitation at around 1400–1700 LST. Precipitation decreases from south to north, indicating a transition from the southern mountainous areas to the IRV and northern mountainous areas. The center of maximum precipitation is in the southern mountainous areas, while the lowest value of 0.05 mm occurs in the morning in the IRV. Figure 9d represents the zonal average. The DVP is greater than the meridional average. Between 80.06°E and 82.08°E, the maximum precipitation, 0.3–0.4 mm, is dispersed. From west to east, precipitation increases and then decreases. Between 82.08°E and 83.03°E, there is a secondary center of precipitation, ranging from 0.15 to 0.2 mm, spanning from 1200 to 1900 LST.

To clearly understand the DVP in the Ili region, it is necessary to understand the spatial pattern, especially in mountainous areas. Johansson and Chen [75] pointed out in their study that, even in mid–latitude regions, the relationship between precipitation and elevation is not a simple linear increase but exhibits significant regional variations. One of the most important processes that takes place is topographic enhancement, which is influenced by factors such as wind speed and direction. In the following, a more comprehensive analysis is conducted on the spatial distribution patterns of diurnal variations in precipitation peaks, and variations in precipitation with respect to zonal, meridional, and elevation changes in the Ili region. Figure 9 shows that topography has a significant impact on the centers with high precipitation values. In order to better analyze the DVCP due to topographical changes, six profiles were selected.

On the southern slope of the Alatau Mountains, at the northern end of the profile, as shown in Figure 10a, the maximum hourly rainfall reaches 0.15 mm. The peak occurs at 1700 LST, and the minimum precipitation value is seen at 0800 LST. From the slope's elevation of 2.2 km to the southern part of the valley, precipitation shows a decreasing trend during different time periods. The rainfall peak occurs at 2300 LST, and the lowest value is found at 1700 LST. In the middle part of the profile, the central part of the Ketman Mountains, the rainfall peak on the northern slope occurs at 1400 LST, while on the southern slope towards the southern end of the profile, the peak occurs at 1700 LST. In Figure 1, the average 750 hPa horizontal wind field during the summer indicates that the southern slope is a lee slope, and the peak precipitation time lags behind the northern slope. On the southern slope of the Haerk Mountains, the maximum hourly rainfall surpasses 0.5 mm, and afternoon and evening hours contain significantly more precipitation than other times of the day. In Figure 10b, the profile terrain resembles that in Figure 10a. The northern part is the southern slope of the Boluokenu Mountains, and the precipitation during some time periods is close to that of the valley. When combined with Figure 1, it can be seen that the average annual precipitation in the mountainous area is much lower than that of mountainous areas at the same latitude. From the transition zone from the valley to the mountainous area, the precipitation peak occurs between 2300 and 0500 LST. The central part is the eastern part of the Ketman Mountains, and the precipitation peak is smaller than the central part (Figure 10a). The Zhaosu Basin, from the depression to the south, experiences peak precipitation slightly later than the surrounding mountainous regions, between 1700 and 2000 LST. The western part of the Haerk Mountains has a lower precipitation peak of 0.34 mm, occurring at 1700 LST, compared to the eastern part. In Figure 10c, the profile is high at both ends and low in the middle. Rainfall peaked in the northern mountainous region at 0.28 mm, and in the southern Nalati Mountain region at 0.35 mm. The peak times in the mountainous areas are all 1700 LST, while the peak precipitation times from the mountainous area to the valley occur sequentially at 1700, 2000, 2300, 0200, and 0500, showing a delayed trend. In the western part of the study area, represented in Figure 10d, it can be seen that the northern mountainous area has an elevation exceeding 3 km, with a rainfall peak of approximately 0.2 mm occurring from

2300 to 0200 LST. This peak time differs significantly from the other mountainous areas. The southern mountainous area has a peak precipitation time of 1400–1700 LST.

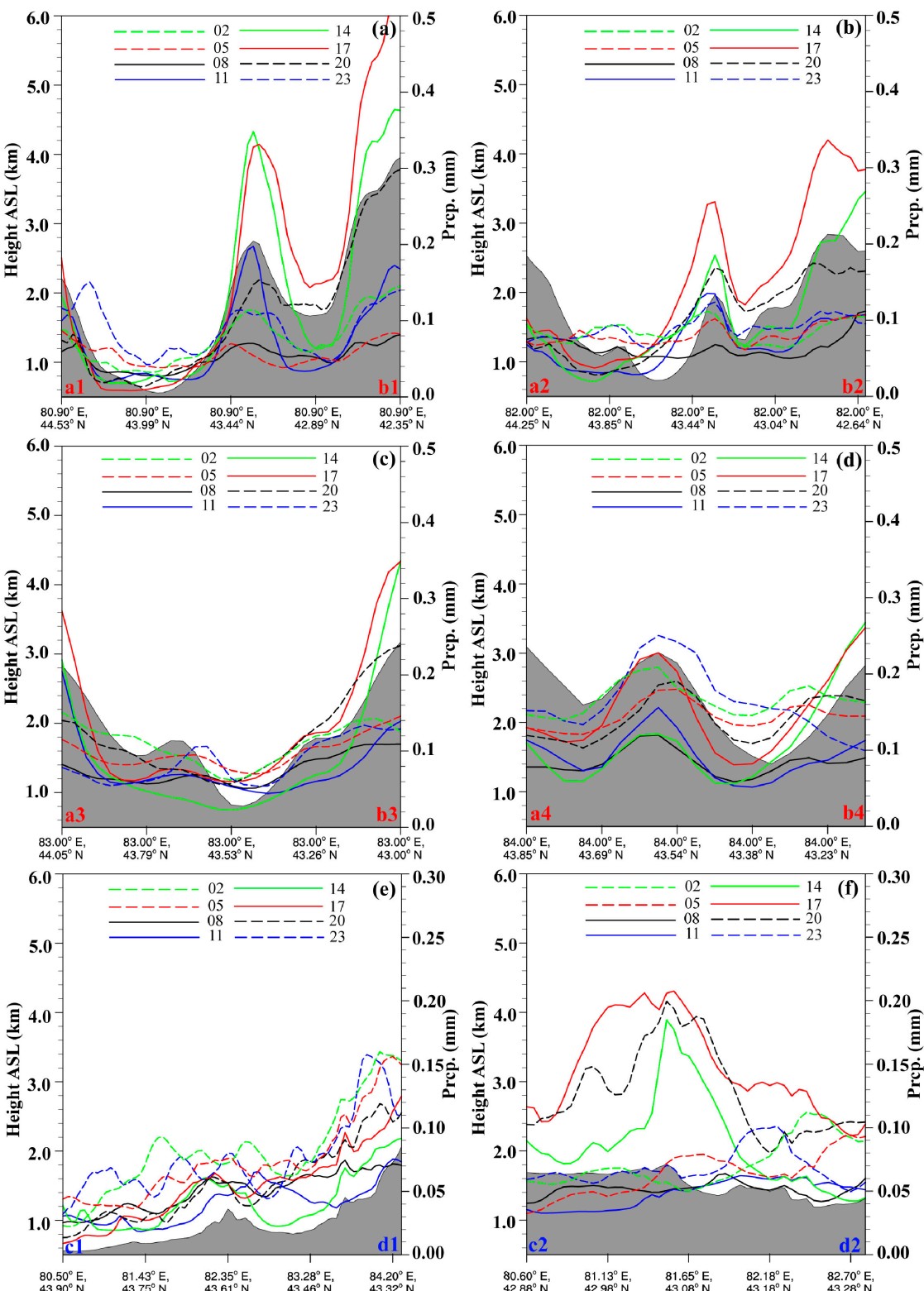

**Figure 10.** During summer (from June to August) in the Ili region from 2015 to 2019, obtained from WRF_NJU data. (**a**–**f**) The spatial distribution of hourly average precipitation every 3 h at different locations along the line segments in Figure 8. The different colored curves represent the PA (mm) corresponding to each time (LST), and the gray shading represents the topography (km).

In Figure 10e, the precipitation profile extends northwest–southeast along the valley. The average elevation from the northwestern region up to the foothills of the Nalati Mountains is below 1 km, and the precipitation during these different time periods is below 0.1 mm. The diurnal variation in precipitation is much smaller than that in the mountainous areas. The peak precipitation occurs from west to east at 2300–0500 LST, and there is a significant difference in the peak time depending on the location. The precipitation peak in the foothills of the Nalati Mountains to the mountain slopes reaches 0.15 mm, and occurs at the same time as that in the valley region. In Figure 10f, the precipitation profile represents a southwest–northeast direction from the depression in the southern mountainous area to the Zhaosu Basin. This figure makes it clear that the Zhaosu Basin's depression has its highest precipitation at about 1700–2000 LST, slightly later than the surrounding mountainous areas. The peak precipitation exceeds 0.2 mm in the basin, while it is approximately 0.1 mm in the central and western parts.

Upon comparing the aforementioned findings, it becomes apparent that the diurnal variations in precipitation near the mountain summits display a considerably greater disparity than those in the slopes, valleys, and basin regions. In the valleys and basins, the hourly precipitation generally remains below 0.1 mm, whereas the slopes observe precipitation levels above 0.1 mm and, near the mountain summits, the precipitation exceeds 0.3 mm. In addition, with regard to the peak precipitation, the influence of terrain and wind patterns results in a slight delay in precipitation on the lee side of certain mountainous areas, compared to the windward side. In contrast, the valleys and basins experience even later peak precipitation, typically occurring between 2300 and 0500 LST.

### 3.3. Statistical Characteristics of Factors Related to Precipitation in the Ili River Valley

3.3.1. The Diurnal Variations in Precipitation and Their Correlation with Different Altitudes

In complex terrain areas, mountains themselves modify local airflow and corresponding weather systems through dynamic and thermal effects, thereby causing localized precipitation and changes in cloud patterns. Existing observational studies have confirmed the close relationship between terrain–induced precipitation and meteorological conditions [11,29,68]. Referring to the previous case study on heavy precipitation and analyzing the statistical characteristics of meteorological factors related to precipitation, we made initial investigations into their contributions to precipitation.

Figure 11a shows the average dispersion profiles plotted at intervals of 0.1 km, from 0.5 to 4.5 km. It clearly illustrates that the dispersion increases with altitude, being below 2 km during the 2000–0800 LST period, with significantly higher dispersion values. There is a large difference in dispersion between the upper and lower levels, and the dispersion values are negative during this period, contributing to low–level convergence and upper–level divergence in the valley and basin areas. From 1100 to 1700 LST, the dispersion values are positive, indicating upper–level divergences in the mountainous regions, which facilitates precipitation and corresponds to the peak precipitation period in the mountainous areas mentioned earlier.

Before and after 0800 LST, the equivalent potential temperature ($\theta e$) from the surface to near 3 km shows an increase with altitude (Figure 11b). Between 1400 and 2000 LST, this phenomenon is found only below 1 km altitude. Around 1700 LST, the highest values of the $\theta e$ occur above 1 km, indicating that the air masses during this period are more heated and contain more energy. Under such conditions, the air masses are relatively unstable and prone to uplift, leading to cloud formation and precipitation.

Except for the 0800 LST profile, the water vapor mixing ratio (WVMR) dramatically drops above 1 km of altitude, indicating a higher water vapor content in the valley region compared to the mountainous areas (Figure 11c). The minimum values occur around 0800 LST, while the maximum values occur around 2000 LST, corresponding to the peak precipitation period in the valley region.

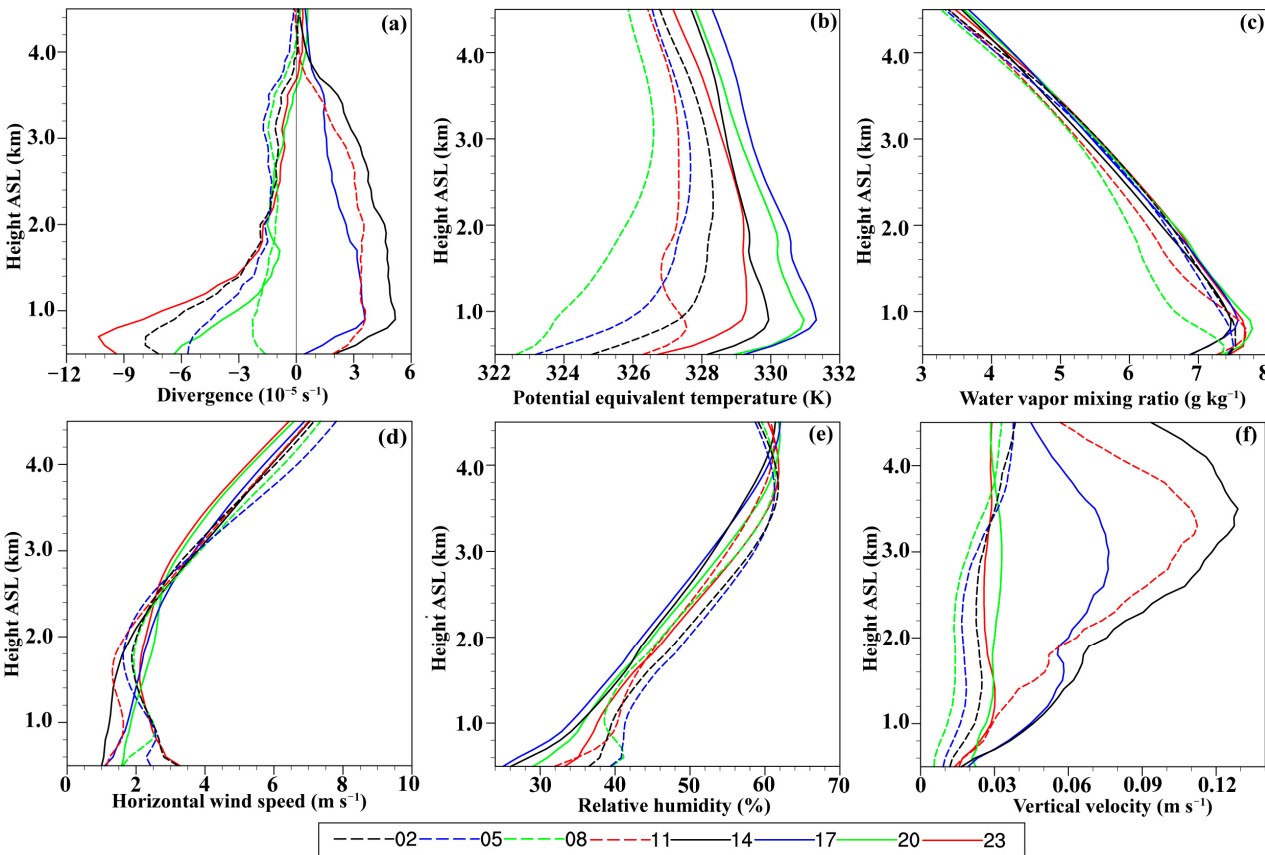

**Figure 11.** During summer (from June to August) in Ili region from 2015 to 2019, obtained from WRF_NJU data. Vertical profiles of mean (**a**) divergence, (**b**) equivalent potential temperature, (**c**) water vapor mixing ratio, (**d**) horizontal wind speed, (**e**) relative humidity, and (**f**) vertical velocity at different times. The vertical profiles depict the variation with height, and the different colored curves represent the changes corresponding to different times from 0200 to 2300 LST.

In Figure 11d, the average horizontal wind speed (HWV) increases significantly with altitude between 1400 and 2000 LST, indicating a rapid increase in HWV that enables the faster movement and accumulation of water vapor from the valley to the mountainous regions. Other times, the HWV exhibits a characteristic of initially increasing and then decreasing between 0.5 and 2 km. The low–level HWV ranges from 1 to 3 m s$^{-1}$, while the upper–level HWV ranges from 6 to 8 m s$^{-1}$.

The relative humidity (RH) increases with altitude from 0.5 to 3.5 km at all time intervals (Figure 11e). The maximum RH exceeds 60% at 3.5 km, indicating a higher RH in the high–altitude areas compared to the valley and basin regions, which is more conducive to saturated precipitation. In the valley and basin regions below 1 km, the RH reaches its maximum value during the dawn period from 0500 to 0800 LST.

In Figure 11f, the vertical velocity (VV) increases rapidly from the surface to 3.5 km between 1100 and 1700 LST, reaching its maximum value around 1400 LST, which coincides with the peak solar radiation during the summer. This increase in solar radiation leads to an increase in surface temperature, causing heated air to ascend. At other times, the VV shows less variation with altitude.

In previous studies, researchers have found that the mechanisms influencing the occurrence and development of heavy precipitation are complex [39,76,77]. In this paper, six relevant meteorological factors are selected. The previous analysis discussed their diurnal variations at different elevations in the Ili region. In order to explore the correlation between meteorological factors at different elevations and precipitation, Figure 12 illustrates the results. Except for some heights of HWV, VV, and WVMR, all other meteorological

factors at different heights pass the significance test at the 0.05 level. Comparatively, the *CC* between $\theta e$ and precipitation is the highest within the range of 1–4.5 km, with all values above 0.88. The *CC* for divergence ranges from 0.44 to 0.83, and shows a strong positive correlation with precipitation. The maximum *CC* appears at an altitude of 4 km, corresponding to low–level convergence and high–level divergence. The *CC* between HWV and precipitation below 1 km ranges from −0.62 to −0.58, indicating a negative correlation. Above 1 km, only the heights of 2 km and 2.5 km pass the significance test, with *CC*s of 0.54 and 0.51, respectively, showing a strong positive correlation. VV passes the significance test at all heights except 4.5 km, and the *CC* decreases with increasing altitude, with a maximum value of 0.88 at 1 km. RH shows a negative correlation below 4 km, with *CC*s ranging from −0.92 to −0.61, except for a positive correlation at 4.5 km with a *CC* of 0.8. The correlation of WVMR exhibits a fluctuating trend of increasing and then decreasing with increasing altitude, with a maximum *CC* of 0.95 occurring at an altitude of 4.5 km.

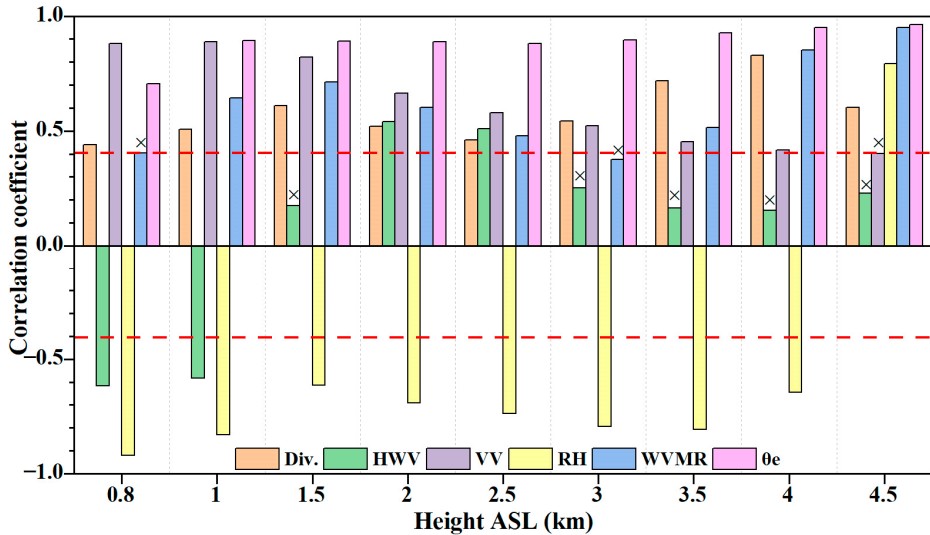

**Figure 12.** During summer (from June to August) in the Ili region from 2015 to 2019, obtained from WRF_NJU data. The correlation coefficients between six variables at different altitude levels and precipitation, with the red dashed line representing the corresponding position for $p \leq 0.05$. $\times$ stands for not passing the test of significance.

Based on the statistical analysis of meteorological factors at different altitudes and their relationship with precipitation, a strong correlation has been found between the six meteorological factors and precipitation. In order to better understand the contributions of various meteorological factors to precipitation in different complex terrain areas, multiple linear regression equations were constructed for five regions identified in Figure 8. The fitted coefficients and adjusted R–squared values presented in Table 3 have passed the significance test at a 95% confidence level. Based on the specific locations and characteristics of the coefficient of variation (*CV*) of precipitation, these regions can be further investigated as high–*CV* and low–*CV* areas.

**Table 3.** The coefficients of multiple linear regression fitting and the adjusted R–squared values.

|  | Div. | $\theta e$ | WVMR | RH | HWV | VV | Adj. $R^2$ |
|---|---|---|---|---|---|---|---|
| WV | −0.0018 | −0.0319 | 0.1718 | −0.0158 | 0.0055 | 0.2606 | 0.7380 |
| NK | −0.0010 | 0.0775 | −0.3964 | 0.0342 | −0.0007 | 0.8899 | 0.7721 |
| EV | −0.0009 | 0.0606 | −0.3145 | 0.0318 | 0.0065 | 0.5321 | 0.8537 |
| KM | 0.0016 | 0.3505 | −1.7853 | 0.1346 | −0.0110 | −0.3946 | 0.9758 |
| HM | −0.0022 | −0.1245 | 1.0494 | −0.0657 | 0.0938 | −0.8355 | 0.9311 |

### 3.3.2. Regression Analysis and Verification of Precipitation Factors in the High–Value Area of *CV*

This region includes WV, KM, and HM. From Figure 13, it can be found that, although all three areas are high–value areas in *CV*, there are significant differences in the temporal distribution and magnitude of precipitation. WV, located in the western part of the valley, exhibits a clear characteristic of more evening rainfall and less afternoon rainfall, which is different from Figure 13b,c. The hourly precipitation is the smallest among the five areas. The temporal distribution of various factors above the surface at 0.5 km reveals a unimodal structure for all six meteorological factors except HWV. Before the peak precipitation in the WV region, HWV and VV increase significantly, indicating an increased upward motion and maximum negative values of divergence, corresponding to low–level convergence. KM and HM, located in mountainous areas, show peak precipitation at 1700 LST and 1600 LST, respectively. HW has more precipitation, and HWV, VV, and divergence, reach their peaks 1–2 h before and after the peak precipitation, indicating favorable dynamic and thermal conditions for precipitation occurrence and development. RH and WVMR also show strong consistency with the daily variation trend of precipitation, providing necessary water vapor for precipitation to occur. According to Table 3, VV makes the largest contribution to the daily variation in precipitation in WV, showing a negative correlation. WVMR makes the largest contribution to the daily variation in precipitation in KW and HW, showing negative and positive correlations, respectively. Comparing the fitted equation results with the precipitation values of WRF_NJU, the scatter points cluster near the 1:1 line. The determination coefficients ($R^2$) for all three areas exceed 0.8; the *RMSE*s are less than 0.03, and *BIAS*s are −0.59%, 0.63%, and 0.29%, respectively. This indicates that the multivariate linear equation results from the three regions have a small deviation from the precipitation values of WRF_NJU and can be used for the prediction of average summer precipitation in this area.

### 3.3.3. Regression Analysis and Verification of Precipitation Factors in the Low–Value Area of *CV*

This region includes the EV and NK areas, as shown in Figure 14a. PA are mostly below 0.1 mm, with the peak occurring from the evening to early morning hours. There are minimal variations in precipitation throughout the day, with the difference between peak and valley values being less than 0.05 mm. HWV reaches its maximum value three hours after the precipitation peak, while VV shows a declining trend before the peak. During the evening, the valley's divergence shows strong negative values, indicating low–level convergence, and there is good consistency between RH, WVMR, and precipitation. In contrast to the mountainous regions, the area shown in Figure 14b is located in the zone from valley to foothills. Precipitation exhibits a distinct bimodal structure, reflecting the diurnal characteristics of both valley and mountain precipitation. The two peaks occur at 1600 and 0100 LST, respectively. HWV rapidly increases before the afternoon precipitation peak, and the precipitation peak aligns with the peaks or secondary peaks in other meteorological factors, showing good consistency. The negative divergence reaches its maximum during the evening peak, aiding in the uplift of convergence in the valley, leading to precipitation formation. Table 3 shows that VV contributes the most to EV and NK, exhibiting a positive correlation with the DVP in that area. In Figure 14c,d, we can observe that the coefficient of determination ($R^2$) exceeds 0.8, the root mean square error is 0.005, and the relative deviation is 1.36% and −1.03%, respectively. These results indicate that the multivariate linear equations predict precipitation, with minimal differences from the reference values, demonstrating good predictive abilities.

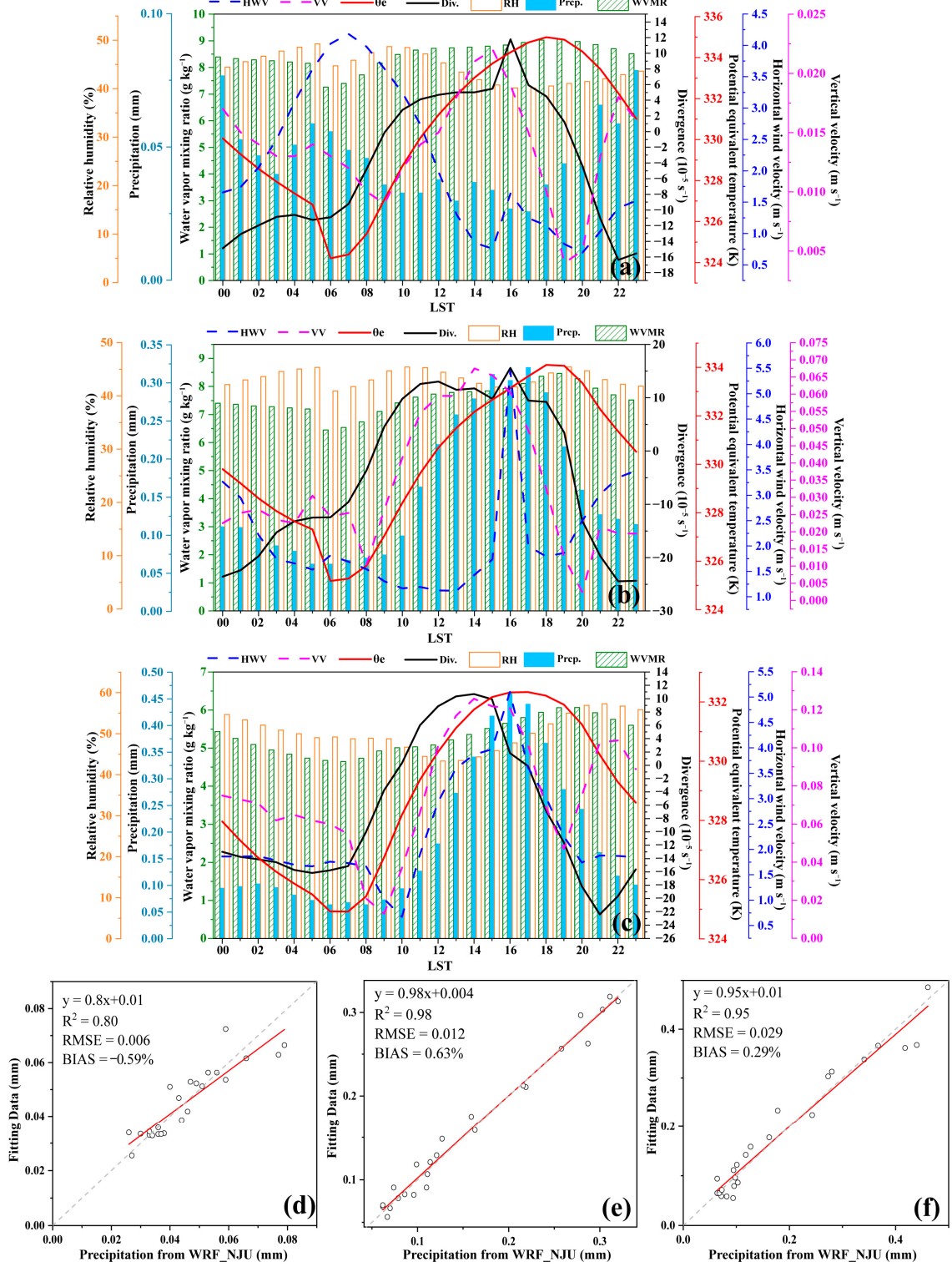

**Figure 13.** During summer (from June to August) in Ili region from 2015 to 2019, obtained from WRF_NJU data. Hourly evolution plots of horizontal wind speed (blue dashed line), vertical velocity (pink dashed line), equivalent potential temperature (red solid line), divergence (black solid line), relative humidity (yellow bars), average precipitation (blue bars), and specific humidity (green bars) for the (**a**) WV (**b**) KM (**c**) HM region in Figure 8. Scatter plot showing a comparison between the simulated precipitation data from the multiple linear regression in the (**d**) WV (**e**) KM (**f**) HM region and the WRF_NJU precipitation data. The red solid line represents the linear fit, and the dashed line represents the 1:1 line.

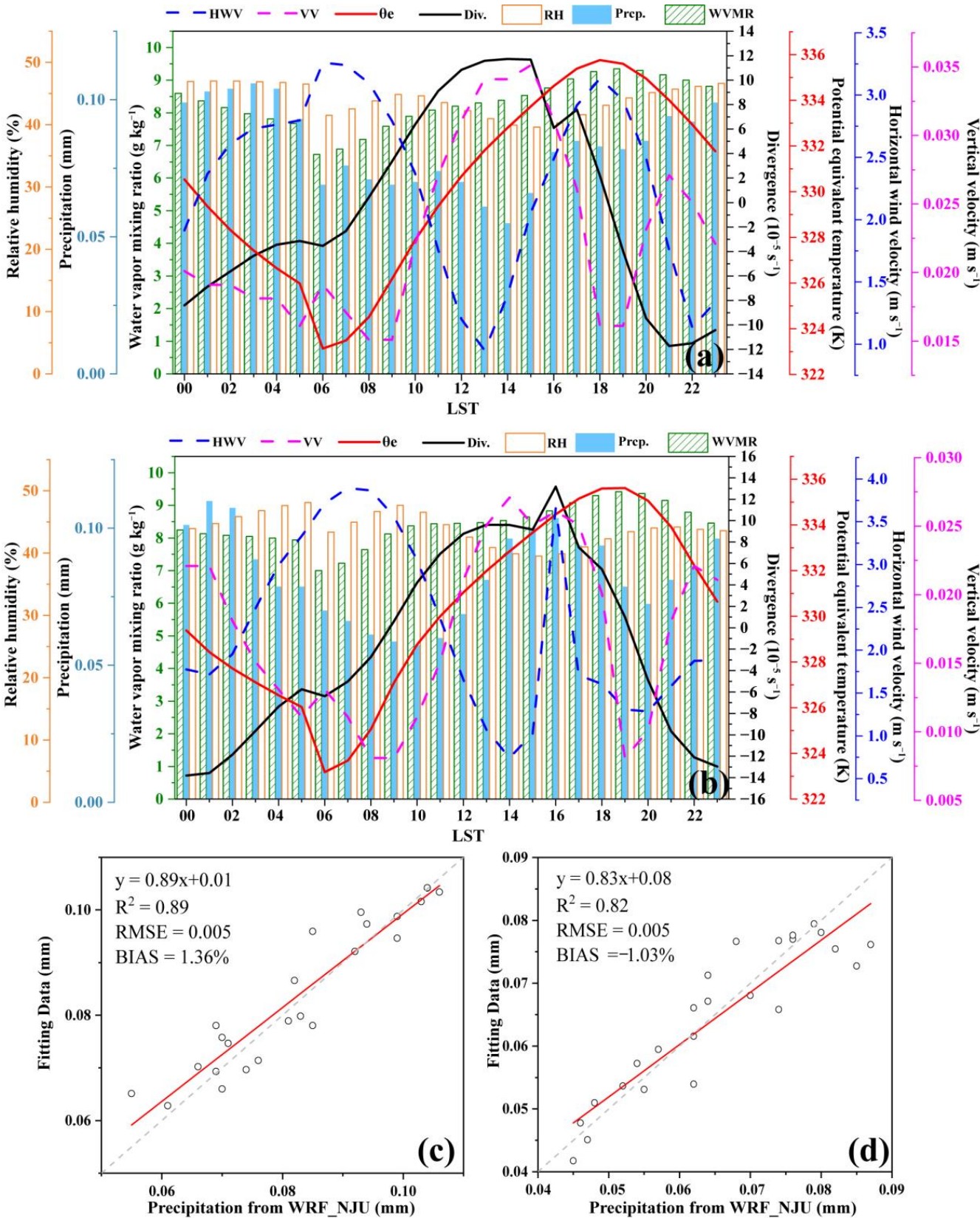

**Figure 14.** Similar to Figure 13, (**a**,**c**) represent EV. (**b**,**d**) represent NK.

In summary, through hourly findings of meteorological factors at different altitudes, we find different diurnal variation characteristics. Except VV, the diurnal variation difference between other factors at low levels is greater than that at high altitudes. By analyzing the correlations between six meteorological factors and precipitation at each altitude, we found significant positive correlations for divergence, VV, and WVMR at all altitudes. Below 1.5 km, HWV shows a negative correlation with precipitation, while above 1.5 km, it shows a positive correlation. RH is negatively correlated with precipitation. By performing multiple regression analyses to determine the contributions of each factor to the DVP, we found significant local variations among different regions. In mountainous areas (KM and HM), WVMR contributes the most to DVP, while in valleys (WV and EV) and slopes (NK), VV contributes the most to DVP. There is a good correspondence between the peak values of DVP and the various meteorological factors. The more complex the terrain, the greater the differences in DVP.

## 4. Discussion

The paper evaluates the applicability of 4 km WRF_NJU data and then conducts a detailed investigation of the DVCP based on the high–tempo–spatial–resolution data from WRF_NJU during the summer of 2015–2019 for the first time in the Ili region, filling the gap left by previous studies due to the lack of high–tempo–spatial–resolution data to understand the DVCP in the Ili region [39,67,68]. The WRF_NJU data can clearly reveal the tempo–spatial evolution characteristics of PA, PF, PI, and peak values in the longitudinal, latitudinal, and altitude dimensions, which can facilitate a detailed analysis of the DVCP in various complex terrains in the region. By utilizing these data, significant differences were found compared to previous studies that only used observations from a few stations to analyze the DVCP [78,79]: the peak values in the eastern valley occurred from the evening to early morning, the peak values in the central valley occurred from the early morning to dawn, and the peak values in the western valley and mountainous areas were found from the afternoon to evening, clearly identifying the regional differences in precipitation peak delays caused by terrain.

Some of the DVCPs in the Ili region are noticeably different from those in other mountainous areas of China, such as Sichuan Basin, although the Ili region is also considered a mountainous region, with surrounding mountains and a rugged terrain [29,44,80]. For instance, in the Ili region, the areas with a high PA, PF, and PI are located in the high–altitude mountainous regions, and there are significant differences between the eastern and western parts of the valley. In contrast, the Sichuan Basin exhibits a precipitation peak center located in the western part of the basin and mountain slopes, with a peak time of around 0200 LST [78]. The mountainous regions have a higher PA compared to the slopes and valleys due to terrain differences, and the timing of precipitation peak occurrence shows a delayed characteristic influenced by the terrain in the Ili region.

Furthermore, the decomposition of precipitation into three major modes using REOF clearly demonstrates the tempo–spatial variation patterns of diurnal changes in the Ili region for the first time and reveals the existence of semi–diurnal structures in the central mountainous and foothill areas. The *CV* results indicate that the areas with high values are located in the mountains, with maximum values exceeding 80%, much higher than in the Sichuan Basin and South China [69], reflecting the instability of weather conditions in arid regions. In addition, this study further investigated the correlation between some basic meteorological factors and diurnal precipitation, and their possible contributions to precipitation, by utilizing the characteristic values of the *CV*. The correlation between meteorological factors and precipitation at various altitudes is mostly in the range of 0.6–0.9, while previous studies in South China reported similar correlation coefficients below 0.5 [77]. Regression analysis revealed that VV and WVMR make significant contributions to the temporal variation in precipitation, providing preliminary insights into the possible mechanisms of precipitation diurnal variation.

This work can be considered a great breakthrough in the research on DVCP in Xinjiang, because previous studies on DVCP in this region were restricted by their low–tempo–spatial–resolution data. Furthermore, many previous studies on precipitation in Xinjiang mainly focused on individual cases of heavy precipitation, which allowed for an in–depth analysis of the mechanisms behind single cases, but lacked generality in their findings.

## 5. Conclusions

This paper evaluates the performance of the WRF_NJU precipitation data and investigates the diurnal variation characteristics of precipitation (DVCP) based on hourly precipitation data from the WRF_NJU during the summer (from June to August) of 2015–2019 in Ili region, Xinjiang, Northwest China. The evaluation includes an analysis of precipitation amount (PA), peak values, and spatial distribution, and utilizes correlation coefficients (*CC*), standard deviations (SD), root mean square errors (*RMSE*), and relative biases (*BIAS*) for comparison. The main conclusions are as follows:

(1) WRF_NJU model effectively reproduces the precipitation in terms of quantity, spatial distribution, and peak values in the Ili region. The PA in the valleys and mountainous areas is close to that of the automatic weather stations (AWS) and Multi–Source Weighted–Ensemble Precipitation (MSWEP) data, but lower than that of ERA5–Land. WRF_NJU exhibits characteristics such as a low *RMSE*, similar SD, high *CC*, and small *BIAS*s throughout the study area. Therefore, WRF_NJU model data can be considered credible data and be used for a statistical analysis of DVCP and the features of related meteorological factors.

(2) The PA, precipitation frequency (PF), and precipitation intensity (PI) exhibit similar diurnal variations. The maximum values occur in the evening near the summit of Ketman Mountains (KM) and Haerk Mountains (HM). The DVCP is more pronounced in the central and southern mountainous areas compared to the valley areas. The average value of moderate to intense precipitation is 1.62 mm h$^{-1}$, contributing to over 80% of the hourly precipitation and total precipitation, with the most prominent values being found from noon to evening.

(3) The precipitation in the Ili region was decomposed into three modes by the Rotated Empirical Orthogonal Function (REOF). Both the first and second modes exhibited distinct differences in diurnal variation peaks between the mountainous areas, valleys, Zhaosu Basin, and wedge–shaped area, attributed to variations in topography. The valley experiences peak precipitation from 2100 to 1100 LST, while the mountainous areas observe peak precipitation from 1200 to 2000 LST. The peak precipitation in the Zhaosu Basin and wedge–shaped area occurs between 1700 and 0200 LST. The third mode reveals semidiurnal variation characteristics near the peak of Ketman Mountain. The *CV* further reflects the maximum differences in DVCP in the western valleys and southwestern mountainous areas of the study area.

(4) The temporal distribution of peak precipitation clearly demonstrates earlier peaks in the mountainous areas compared to the valleys, with the western segment of the river valley experiencing earlier peaks than the eastern segment. Along the longitudinal and latitudinal directions, as well as in terms of elevation, the mountainous areas exhibit greater precipitation than the valleys, probably due to the enhanced topographical forcing. Precipitation peaks slightly earlier on downwind slopes than on upwind slopes. The differences in precipitation amount gradually increase from valleys, foothills, slopes, to mountain peaks. The peak precipitation on slopes and mountain peaks occurs from the afternoon to evening, while in valleys, it appears from late night to early morning. Foothills exhibit a bimodal pattern. Zhaosu Basin, despite its low topography, exhibits diurnal variation characteristics similar to mountainous areas due to its higher elevation.

(5)　The precipitation is influenced by multiple meteorological factors, and there is a strong correlation between meteorological factors and precipitation at different heights and different times. The multiple linear regression equation effectively captures the diurnal variation characteristics of summer precipitation in various places in the Ili region. In the areas with high values and low values of the coefficient of variation (*CV*), water vapor mixing ratio (WVMR) and vertical velocity (VV) contribute significantly to precipitation. The WVMR (VV) seemed to play a more significant role in mountainous (valleys) areas. The meteorological factors within the region exhibit a good correspondence with the peak values of precipitation, as they change over time.

However, the understanding of complex terrains, local valley wind circulation, precipitation propagation mechanisms, weather circulation background, and the formation mechanism of DVCP in this region remains unclear. Further research is needed to investigate the relationship between DVCP and large–scale circulation patterns, complex terrain, and its dynamic and thermal mechanisms, as well as other influencing factors in the Ili region.

**Author Contributions:** Data curation: Z.L. and K.Z.; formal analysis: Z.L. and Z.K.; investigation: Z.L., A.A. (Abuduwaili Abulikemu), and A.A. (Abidan Abuduaini); methodology: Z.L. and A.A. (Abuduwaili Abulikemu); project administration: A.A. (Abuduwaili Abulikemu); resources: K.Z., Z.L., J.L. and Y.Z.; software: Z.L., K.Z., C.L. and Q.S.; supervision: A.A. (Abuduwaili Abulikemu); validation: Z.L. and A.A. (Abuduwaili Abulikemu); writing—original draft: Z.L., A.A. (Abuduwaili Abulikemu), and A.A. (Aerzuna Abulimiti); writing—review and editing: Z.L., A.M. and A.A. (Abuduwaili Abulikemu). All authors have read and agreed to the published version of the manuscript.

**Funding:** This work was sponsored by the National Natural Science Foundation of China (42265003), Natural Science Foundation of Xinjiang Uygur Autonomous Region (2022D01C359), Scientific and Technological Innovation Team (Tianshan Innovation Team) project (Grant No. 2022TSYCTD0007), The Sub–project of the Third Xinjiang Scientific Expedition (2022xjkk030502), Natural Science Foundation of China (U2003106), National Key Research and Development Program of China (2018YFC1507103), 100 Young Doctors Introduction Program of Xinjiang (Tianchi Doctor Program) Foundation (50500/04231200737), Doctoral Research Startup Foundation of Xinjiang University (50500/62031224618).

**Data Availability Statement:** The hourly surface automatic observation station dataset was collected and compiled by the China Meteorological Administration (CMA);WRF_NJU provided by Key Laboratory of Mesoscale Severe Weather of Nanjing University; The ERA5–Land reanalysis dataset from the European Center for Medium–Range Weather Forecasts (ECMWF) was used as auxiliary data in this study(https://cds.climate.copernicus.eu/cdsapp#!/dataset/reanalysis-era5-land?tab=form, accessed on 15 December 2022); MSWEP data could obtain the public application (https://www.gloh2o.org/mswep/, accessed on 10 January 2023).

**Acknowledgments:** We thank the four anonymous reviewers and all editors for their valuable comments, suggestions and efforts during the handling of our manuscript. We also thank for the High–Performance Computing Center of Nanjing University for conducting the numerical calculations in this paper on their IBM Blade cluster system.

**Conflicts of Interest:** The authors declare no conflict of interest.

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
