# Peer review of "Diurnal Variation Characteristics of Summer Precipitation and Related Statistical Analysis in the Ili Region, Xinjiang, Northwest China"

_remotesensing, doi:10.3390/rs15163954_

Round 1

Reviewer 1 Report

The present manuscript, entitled “Diurnal variation characteristics of summer precipitation and 2 related statistical analysis in the Ili region, Xinjiang, Northwest 3 China ” by Li et al., presents a comprehensive study on the diurnal variation characteristics and statistical features of summer precipitation in Ili region using the 4-km WRF simulation. I find this paper interesting, and it ranks well in the most recent literature on applications with regional climate models. The paper is quite well written, although the manuscript needs some revision before being considered for publication.

1. Line 47-48: This research article should be cited when you discuss about the fact that the intensity and frequency of extreme rainfall events are increasing in many parts of the world with global climate warming.

Shahi, N. K., Rai, S., Verma, S., & Bhatla, R. (2023). Assessment of future changes in high-impact precipitation events for India using CMIP6 models. Theoretical and Applied Climatology, 151(1-2), 843-857.

Also include the discussion about the fact that the future projections of global and regional monsoons are still quite uncertain in terms of precipitation estimates and require more careful investigation (IPCC 2013).

2. A table with model description will be beneficial for readers. So, it should be included.

3. The visibility of all figures should be improved. It is very difficult to see it in detail.

4. It is also important to include a discussion about the observational uncertainty over the study area. This is an important aspect, especially when you discuss about extreme hourly precipitation, also any overestimation of the extremes by the model might not be necessarily wrong (but only under represented in observations)

Prein, A. F., & Gobiet, A. (2017). Impacts of uncertainties in European gridded precipitation observations on regional climate analysis. International Journal of Climatology, 37(1), 305-327.

5. Author should mention some studies (if any) regarding the impact of climate change on agriculture activities/production across the study region to strengthen the importance of this study.

6. Did the author apply REOF analysis on seasonal values of each hour or any other ways?

7. Line 748-749: “mountainous areas exhibit greater precipitation than the valleys.” this could be due to the enhanced topographical forcing

Minor revision is required

Reviewer 2 Report

This paper evaluates the performance of the WRF-NJU and investigates the diurnal variations of precipitation in Ili region. Some major modifications are needed to refine this manuscript for more meaningful outcomes.  Therefore, I suggest that major modifications have to be done:

1.     The paper is submitted to Remote Sensing, presumably because some of the used data stem from satellite missions. This is far from being enough to justify that the paper fits to the scope of Remote Sensing. The authors need to provide some justification for their choice. 

2.     For the data, why do you use MSWEP and ERA-land? These datasets have a corase resolution of 0.1o for the small area studies. How to resolve the uncertainties of these datasets and how to deal the scale mismatch.

3.     The influence factors are based on the correlation coefficient. Why not use the spearman correlation coefficient? Please make some explanation for this.

4.     The figures are not clear enough. The capital of Figure 14 is missing. The number of the figure is much, I suggest delete some.

5.     Some sentences are expressed inaccurately and need to be rewritten, such as line 128: The model WRF-ARW was not developed by Nanjing University.

This paper evaluates the performance of the WRF-NJU and investigates the diurnal variations of precipitation in Ili region. Some major modifications are needed to refine this manuscript for more meaningful outcomes.  Therefore, I suggest that major modifications have to be done:

1.     The paper is submitted to Remote Sensing, presumably because some of the used data stem from satellite missions. This is far from being enough to justify that the paper fits to the scope of Remote Sensing. The authors need to provide some justification for their choice. 

2.     For the data, why do you use MSWEP and ERA-land? These datasets have a corase resolution of 0.1o for the small area studies. How to resolve the uncertainties of these datasets and how to deal the scale mismatch.

3.     The influence factors are based on the correlation coefficient. Why not use the spearman correlation coefficient? Please make some explanation for this.

4.     The figures are not clear enough. The capital of Figure 14 is missing. The number of the figure is much, I suggest delete some.

5.     Some sentences are expressed inaccurately and need to be rewritten, such as line 128: The model WRF-ARW was not developed by Nanjing University.

Reviewer 3 Report

General comments:

This manuscript studies the diurnal variation characteristics and basic statistical features of summer precipitation in Ili region from 2015 to 2019 based on 4-km resolution WRF model simulation data. In particular, the authors identified some significant diurnal variation characteristics of precipitation amount (PA), precipitation frequency (PF), precipitation intensity (PI), and they also conducted some related statistical analysis such as REOF (Rotated Empirical Orthogonal Function) and CV (coefficient of variation) in order to further investigate some related features based on the WRF model simulation data. Due to the relatively sparse population, lagged economic development, complex terrain and harsh environment such as desert and gobi, the number or density of conventional and automatic weather stations in Xinjiang are much less than that of mid-eastern part of China. Therefore, the 1-hr and 4-km resolution simulation data with consecutive 5-year time range can undoubtedly provide a good opportunity to investigate much more detailed characteristics of diurnal variation of precipitation in the region. Therefore, I consider this study as an important contribution to the topic of diurnal variation characteristics of precipitation in Xinjiang, and I recommend accepting this manuscript after addressing minor revision suggestions.

Minor comments:

1. Line 41, “in both high- and low-value of coefficient of variation areas” should be revised to “in both areas with high- and low-value of coefficient of variation”.

2. Line 42, “seemed to play more significant role” should be revised to “probably had played more significant role”.

3. Line 43, the term “Ili River Valley” should be revised to “Ili region”.

4. Lines 93,94, the quotations are not specific enough, please add some proper introductory information.

5. Lines 96-98, the diurnal variation characteristics of precipitation in complex terrain or mountainous areas have been mentioned earlier in the manuscript, therefore it is unnecessary to repeat this content, so I recommend deleting this content.

6. Line 176, What is the consideration of using the ERA5-Land reanalysis data? Please add some explanation in the manuscript.  

7. Lines 194, 206, 215, 221, 222, 223, the format of those formulas looks irregular.

8. Line 356, This sentence seemed to “too absolute” from the perspective of describing a fact based on other’s conclusions. I recommend to add at least one more citation here, and revise this sentence as “According to previous studies [67-XX], the DVP is usually determined by the PF and PI.”

9. Line 359, the term “in the study area” is redundant here, because this term is just mentioned in the previous sentence, so it is unnecessary to repeat and emphasize “the study area” again. Similar repeated terms can be found in lines 435, 450, 452, and 570, and I suggest to remove these replaces terms “in the study area”.

10. Line 398, the “edge-shaped area” mentioned in this manuscript (also mentioned in line 38 in the Abstract and lines 738 and 741 in the “Conclusions” section) is not clear for readers. Please give a proper explanation of the “edge-shaped area” when it occurred for the first time in the main part of the manuscript.

11. Lines 446-448, I suggest to revise this sentence as “The WRF_NJU data with relatively high tempo-spatial resolution can provide a good opportunity to investigate much more detailed characteristics of DVCP in Xinjiang, because the density of distribution of the conventional and automatic weather stations in Xinjiang are much less than that of mid-eastern part of China, owing to the relatively sparse population, lagged economic development, complex terrain and harsh environment such as desert and gobi.”.

12. Lines 448-449, “Figure 9a shows the spatial distribution of peak PA in summer.” would be better to revised to “Figure 9a shows the spatial distribution of the diurnal variation of the peak time of PA which obtained from the WRF_NJU data.”

13. Line 488, the time-longitude hovmöller diagram (Figure 9c, 9e) and time-latitude hovmöller diagram (Figure 9d, 9f) are shown for the Ili region and the red box, respectively. Considering that there are little differences in the tempo-spatial distributions characteristics of the precipitation between those two kinds of figures, I suggest to keep only one of them in the manuscript. Because, the study area mentioned in the title of the paper is “Ili region”, I think the figures calculated in the Ili region (i.e., figs 9c and 9d) should be retained. Besides, the units of the altitudes in all figures should be unified into “km”.

14. Lines 489-490, please add some detailed information in the figure captions such as “during summer (from Jun. to Aug.) in Ili region from 2015 to 2019 obtained from WRF_NJU data”. Please add similar information in other similar figure captions.

15. Line 498, There are some ambiguities in the references cited here, so it is suggested to restate them.

16. Lines 547 and 700, the term “observations” here is used by mistake. All of the related analysis were conducted based on the WRF simulation data, not observation. Please check the term “observation” throughout the paper.

17. Lines 561-604, the unit "m" used in this section should be in consistent with the related content in the figure. I recommend to change it to "km".

18. Line 566, the mark “c1” at the lower left corner of the Figure 10e should be labelled in the figure, rather than outside the figure, for the sake of in consistent with other panels in Figure 10.

19. Lines 578,581, The format of the “ is incorrect.

20. Line 605, What is the basis for selecting and analyzing these 6 meteorological factors in Figure 11? What are the considerations in selecting relative humidity and water vapor mixing ratio? Please add some related explanations in the manuscript.

21. Line 630, some bars in the Figure 12 that do not passed the significance test (i.e., below the red dashed line) are not obvious, and I recommend special labeling for them.

22. Line 630, the numbers in the horizontal axes of the Figure 12 should be labelled in the outside of the figure along with all tick marks.

23. Line 701, the term "their own diurnal variations" cannot accurately describe the differences between the meteorological factors, I suggest rephrasing contents about the “diurnal variations”.

24. All employed mathematical expressions can be simplified

25.The quality of all diagrams must be improved in the revision.

 Moderate editing of English language required

Reviewer 4 Report

This manuscript investigates the diurnal variation characteristics and basic statistical features of summer precipitation in the Ili region from June to August, between 2015 and 2019. The analysis is based on 4-km resolution Weather Research and Forecasting (WRF) model simulation data from Nanjing University (WRF_NJU). The study is interesting and meaningful to readers, and the manuscript is well-organized and logical. I have two suggestions for the authors:

  1. In the methods section, it is mentioned that the analysis is conducted using 4-km resolution WRF model simulation data. However, it is recommended to provide more detailed information regarding the time range, spatial extent, and resolution of the data. Additionally, including the configuration and parameter settings of the WRF model would enhance readers' understanding of the research methodology and data reliability. The current description is limited, and providing more specifics would be beneficial.

  2. It is suggested to adjust the order of the discussion and conclusion sections to improve the overall flow and logical progression of the manuscript. Reorganizing these sections would enhance readability and ensure a more comfortable reading experience for the audience.

I hope these reviewer's comments will be helpful for polishing your rainfall article. Best of luck with your submission! Feel free to ask if you have any further questions.

Round 2

Reviewer 2 Report

I have no other suggestions for this study.

Reviewer 4 Report

Accept in present form